# The epidemicity index of recurrent SARS-CoV-2 infections

Lorenzo Mari [1✉], Renato Casagrandi [1], Enrico Bertuzzo [2], Damiano Pasetto [2], Stefano Miccoli [3], Andrea Rinaldo [4,5✉] & Marino Gatto [1✉]

Several indices can predict the long-term fate of emerging infectious diseases and the effect of their containment measures, including a variety of reproduction numbers (e.g. $\mathcal{R}_0$). Other indices evaluate the potential for transient increases of epidemics eventually doomed to disappearance, based on generalized reactivity analysis. They identify conditions for perturbations to a stable disease-free equilibrium ($\mathcal{R}_0 < 1$) to grow, possibly causing significant damage. Here, we introduce the epidemicity index $e_0$, a threshold-type indicator: if $e_0 > 0$, initial foci may cause infection peaks even if $\mathcal{R}_0 < 1$. Therefore, effective containment measures should achieve a negative epidemicity index. We use spatially explicit models to rank containment measures for projected evolutions of the ongoing pandemic in Italy. There, we show that, while the effective reproduction number was below one for a sizable timespan, epidemicity remained positive, allowing recurrent infection flare-ups well before the major epidemic rebounding observed in the fall.

[1] Dipartimento di Elettronica, Informazione e Bioingegneria, Politecnico di Milano, Milano, Italy. [2] Dipartimento di Scienze Ambientali, Informatica e Statistica, Università Ca' Foscari Venezia, Venice, Italy. [3] Dipartimento di Meccanica, Politecnico di Milano, Milano, Italy. [4] Laboratory of Ecohydrology, École Polytechnique Fédérale de Lausanne, Lausanne, Switzerland. [5] Dipartimento ICEA, Università di Padova, Padua, Italy. ✉email: lorenzo.mari@polimi.it; andrea.rinaldo@epfl.ch; marino.gatto@polimi.it

The current COVID-19 pandemic calls for modeling tools to assess emerging disease features[1–4], containment measures[5–10] avoiding social and economic havoc[11–13], and the prevention of epidemic rebounding[3,9,14]. To tackle these problems, a common goal is to control the basic reproduction number $\mathcal{R}_0$[15–21] down to values $\mathcal{R}_c$ (control reproduction number) smaller than unity. However, even diseases with $\mathcal{R}_c < 1$ may actually exhibit epidemic phases, although no asymptotic endemism is established[22]. Stricter conditions are required to avoid these subthreshold epidemics[23]. They are based on the concept of epidemicity, akin to that of generalized reactivity in dynamical systems[24,25]. Generalized reactivity allows determining whether and under which conditions impulsive perturbations to a stable steady state can be amplified by the dynamics of the system before eventually fading out. Similarly, epidemicity analysis can help define necessary conditions for transient epidemic outbreaks to occur in epidemiological systems characterized by $\mathcal{R}_0$ (or $\mathcal{R}_c$) < 1.

An epidemicity index can be defined and evaluated for any compartmental epidemiological model described as a set of ordinary differential equations, independently of the disease being addressed, the relevant route(s) of transmission, and the complexity of the (possibly spatially explicit) contact network—as shown by previous applications to spatially implicit or explicit models for vector-borne or water-related diseases[26,27]. Here, we develop the epidemicity index ($e_0$ or $e_c$, for uncontrolled vs. controlled disease spread) for COVID-19, a threshold-type metric defined as the spectral abscissa of a matrix describing the short-term reactivity properties of a compartmental model for COVID-19 transmission. Differently from incidence-based approaches, which typically aim to interpret[28,29] or anticipate[30–32] the time course of an epidemic, epidemicity indices cannot be used to quantitatively predict the occurrence, size, or timing of a specific subthreshold outbreak. Rather, a positive epidemicity index represents a warning about the potential for the epidemic dynamics to surge via coalescence of subthreshold flares; by contrast, a negative epidemicity index structurally rules out any possible occurrence of subthreshold outbreaks. Therefore, control strategies should precautionarily be designed to achieve a negative value of the epidemicity index in order to prevent epidemic rebounding.

To properly estimate reproduction numbers and epidemicity indices in a spatially explicit setting, a natural framework to describe epidemic dynamics unfolding over real or realistic landscapes, it is fundamental to account for spatial connectivity, because estimates of the reproduction number and epidemicity index evaluated at local (i.e., via homogeneous-mixing models) vs. spatial scales (e.g., via metapopulation models) can diverge[27,33,34]. Such connectivity, depending on the spatial scale of interest, can be induced e.g., by the mobility of human hosts[4–6] or by the movement of pathogen receptacles like droplets charged with SARS-CoV-2 viral loads[35,36]. Here, we consider a large-scale metapopulation model describing a set ($i = 1, n$) of communities with baseline population $N_i$ connected by human mobility, subdivided into the COVID-19-relevant epidemiological compartments (Table 1) of susceptible ($S_i$), exposed ($E_i$), post-latent infectious (also termed pre-symptomatic, $P_i$), symptomatic infectious ($I_i$), asymptomatic infectious ($A_i$, including paucisymptomatic), and recovered individuals ($R_i$) in each community $i$ (SEPIAR model[7,14]; see "Methods" and Supplementary Fig. 1). In SEPIAR, the force of infection[7,14], $\lambda_i$, accounts not only for locally acquired infections, but also for those caused by movement of susceptible and infectious individuals. The model includes three types of containment measures: reduction of social contacts, use of personal protection equipment, and/or local mobility restrictions (all subsumed by the percent reduction in transmission rates, $\epsilon_i$); travel restrictions between communities $i$ and $j$ ($\xi_{ij}$); and isolation

of individuals of infected compartment $X$ (rates $\chi_i^X$, $X \in \{E, P, I, A\}$ [$T^{-1}$]). Individuals removed from the community (either hospitalized or quarantined) are also tracked in SEPIAR.

## Results

**Generalized reproduction numbers and epidemicity indices.** We establish conditions for possible long-term circulation of the pathogen in a naïve population (i.e., lacking any prior immunity), either in uncontrolled settings ($\mathcal{R}_0$) or when containment efforts are instituted ($\mathcal{R}_c$)[19,21] via a next-generation matrix (NGM) approach[22,33,34,37] ("Methods"). $\mathcal{R}_0$ and $\mathcal{R}_c$ are the spectral radii $\rho(\cdot)$ of generalized reproduction matrices[7,33,34] accounting for local transmission and mobility fluxes. Specifically, in the absence of controls we obtain:

$$\mathcal{R}_0 = \rho(\delta^E[\boldsymbol{\theta}^P + \sigma\delta^P\boldsymbol{\theta}^I(\boldsymbol{\phi}^I)^{-1} + (1-\sigma)\delta^P\boldsymbol{\theta}^A(\boldsymbol{\phi}^A)^{-1}](\boldsymbol{\phi}^E\boldsymbol{\phi}^P)^{-1}),$$
$$(1)$$

where $\delta^{E,P}$ are the rates at which exposed individuals enter the post-latent stage or post-latent individuals develop either symptomatic (a fraction $\sigma$) or asymptomatic infections; $\boldsymbol{\theta}^X$ are matrices describing disease transmission from the three infectious classes ($X \in \{P, I, A\}$), incorporating mobility; $\boldsymbol{\phi}^X$ are diagonal matrices representing the transition rates from the infection subsystem ($X \in \{E, P, I, A\}$) to the other compartments of SEPIAR ("Methods"). The $\boldsymbol{\theta}^X$ matrices change because of control measures implementing contact reductions ($0 \leq \epsilon_i \leq 1$) and travel restrictions ($0 \leq \xi_{ij} \leq 1$), while the $\boldsymbol{\phi}^X$ matrices change owing to isolation of infected individuals (at rates $\chi_i^X$). Thus, the control reproduction number is $\mathcal{R}_c = \mathcal{R}_0(\boldsymbol{\theta}_c^P, \boldsymbol{\theta}_c^P, \boldsymbol{\theta}_c^A, \boldsymbol{\phi}_c^E, \boldsymbol{\phi}_c^P)$, where $\mathcal{R}_0(\cdot)$ is the same functional relation of Eq. (1), and the matrices $\boldsymbol{\theta}_c^X$ and $\boldsymbol{\phi}_c^X$ are the counterparts of $\boldsymbol{\theta}^X$ and $\boldsymbol{\phi}^X$ ("Methods").

Generalized reactivity[24,25] can be used to determine the occurrence of subthreshold epidemics, namely whether perturbations to a stable equilibrium of SEPIAR may determine a temporary increase of the Euclidean norm $||\mathbf{y}||$ of a suitable system output vector $\mathbf{y}$ obtained by a linear transformation of the infection subsystem ("Methods"). Given the emerging nature of SARS-CoV-2, we focus only on transient dynamics associated with perturbations to the disease-free equilibrium (DFE). The epidemicity indices, $e_0$ (uncontrolled disease spread) and $e_c$ (with containment measures), are the spectral abscissae $\Lambda_{max}^{Re}(\cdot)$ of the Hermitian matrices $\mathbf{H_0}$ and $\mathbf{H_c}$ ("Methods"), constructed from SEPIAR and the structure of the output. Short-term flare-ups are possible only if

$$e_0 = \Lambda_{max}^{Re}(\mathbf{H_0}) > 0 \quad \text{or} \quad e_c = \Lambda_{max}^{Re}(\mathbf{H_c}) > 0 \qquad (2)$$

in the cases of basic or control epidemicity indices ("Methods"). Here the spectral abscissa is defined as the largest real part of the eigenvalues of a matrix (note the difference with the spectral radius, which is the largest module of the eigenvalues).

To illustrate the above concepts, Fig. 1 shows results for a two-community implementation of SEPIAR, in which $\mathbf{y}$ is the eight-dimensional vector with components $E_{1,2}, P_{1,2}, I_{1,2}, A_{1,2}$. If $\mathcal{R}_0 < 1$ and $e_0 < 0$ (Fig. 1a–c), system trajectories (a) converge rapidly toward the DFE ($\mathcal{R}_0 < 1$), and both total prevalence of infection (b) and system output (c) decline monotonically over time in the system output ($e_0 < 0$). If $\mathcal{R}_0 < 1$ and $e_0 > 0$ (d–f), the initial focus of infections causes a noticeable increase in total disease prevalence. Trajectories eventually converge to the DFE, although over a much longer timescale: the total duration of the infection increases from about 50 days in Fig. 1b to more than 150 days in (e). Also, depending on initial conditions, the system output may either monotonically decrease after a perturbation (with small rebounds being possible later on) or show a transitory increase. The latter behavior represents the signature of positive

**Table 1 State variables of model (Eq. (3)).**

| Variable | Definition |
| --- | --- |
| $S_i$ | Susceptible individuals in community $i$ |
| $E_i$ | Exposed (latently infected) individuals in community $i$ |
| $E_i^q$ | Exposed individuals from community $i$ who are quarantine at home |
| $P_i$ | Post-latent (incubating infectious) individuals in community $i$ |
| $P_i^q$ | Post-latent individuals from community $i$ who are quarantined at home |
| $I_i$ | Symptomatic infectious individuals in community $i$ |
| $I_i^h$ | Symptomatic individuals from community $i$ who are treated at a hospital |
| $A_i$ | Asymptomatic or paucisymptomatic infectious individuals in community $i$ |
| $A_i^q$ | Asymptomatic or paucisymptomatic individuals from community $i$ who are quarantined at home |
| $R_i$ | Recovered individuals in community $i$ |

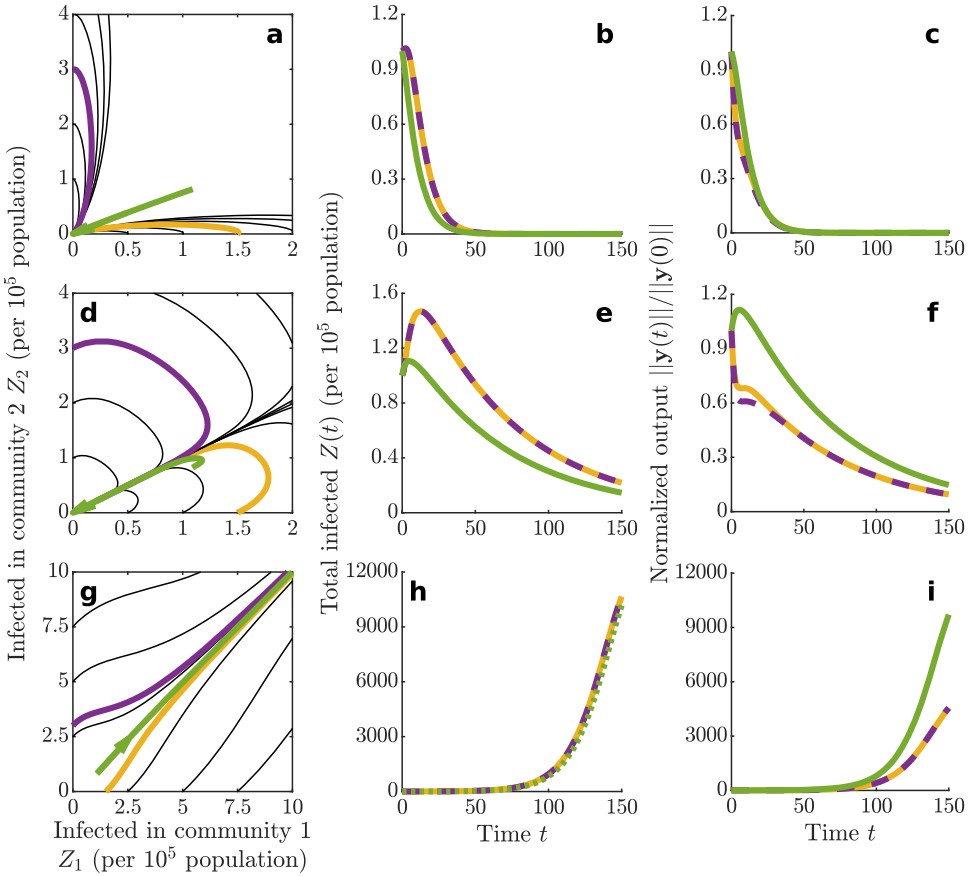

**Fig. 1 A graphical illustration of the concept of epidemicity.** Computational results for COVID-19 transmission in two human communities connected by mobility. **a** System trajectories projected onto the plane spanning the total number of infected people in each of the two communities ($Z_i = E_i + P_i + I_i + A_i$, $i = 1, 2$). Trajectories have been initialized with a few exposed individuals in either community (black, yellow, and purple curves), or with a mix of infected individuals in both communities (green, corresponding to the perturbation of the DFE with the fastest growth in the system output[25]). For this parameter combination ($\beta_1^P = \beta_2^P = 1.2 \times 10^{-1}$ days$^{-1}$), all trajectories converge to the DFE ($\mathcal{R}_0 < 1$, $e_0 < 0$). **b** Temporal dynamics of the total number of infected people in the two communities ($Z(t) = \sum_i [E_i(t) + P_i(t) + I_i(t) + A_i(t)]$). Transmission chains fueled by the initial seeding of infected people decline rapidly over time. **c** Temporal dynamics of the system output, defined as the Euclidean norm of the vector whose components correspond to the infection subsystem ($w^X = 1$, with $X \in \{E, P, I, A\}$). All trajectories are characterized by a monotonic decline in the system output. **d–f** As in (**a–c**), for a parameter combination ($\beta_1^P = \beta_2^P = 4.2 \times 10^{-1}$ days$^{-1}$) resulting in $\mathcal{R}_0 < 1$ and $e_0 > 0$. In this case too, all trajectories converge to the DFE (**d**), but disease prevalence exhibits a peak, later declining slowly over time (**e**). Also, for suitable initial conditions, a transitory increase of the system output following a pulse perturbation is possible (**f**). **g–i** As in (**a–c**), for a parameter combination ($\beta_1^P = \beta_2^P = 7.0 \times 10^{-1}$ days$^{-1}$) resulting in $\mathcal{R}_0 > 1$ and $e_0 > 0$. In this case, trajectories exponentially diverge from the DFE (**g**), and a large outbreak is observed in both disease prevalence (**h**) and the system output (**i**). In these examples, the population size of the first community ($N_1 = 10^6$) is twice as large as the size of the second ($N_2 = 5 \times 10^5$), and the people of the first community are less mobile than those of the second ($M_{12}^{S,E,P,A} = 1/10$, $M_{21}^{S,E,P,A} = 2/3$; symptomatic individuals are assumed not to move from either community, $M_{12}^I = M_{21}^I = 0$). See "Methods" for details and Table 2 for other parameters.

epidemicity ($e_0 > 0$), by which transient epidemic flare-ups are possible even when $\mathcal{R}_0 < 1$. Similar results are obtained for spatially heterogeneous values of the local basic reproduction

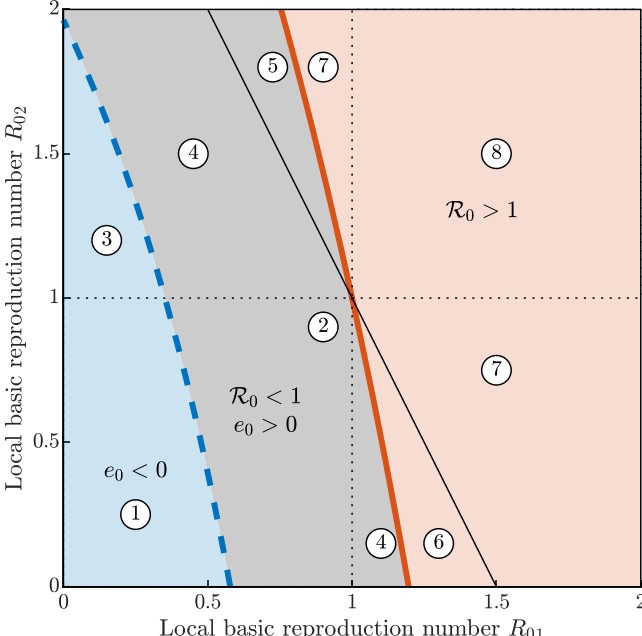

**Fig. 2 The inadequacy of local reproduction numbers in spatially explicit systems.** A catalog of dynamical behaviors for the two-community system of Fig. 1 (red: large outbreaks and endemic transmission are possible because $\mathcal{R}_0 > 1$; gray: transient epidemic waves may be possible because $e_0 > 0$, but transmission fades away eventually because $\mathcal{R}_0 < 1$; blue: no epidemic waves because $e_0 < 0$). See text for an explanation of the specific dynamical outcomes in different parameter subregions (① to ⑧). Note that points ①, ②, and ⑧ correspond to the parameter combination used in Fig. 1 for (a–c), (d–f), and (g–i), respectively. The two communities are characterized by different values of the local basic reproduction number $R_{0i} = \frac{\delta^E}{\mu + \delta^E} \frac{\delta^P}{\mu + \delta^P} \left[ \frac{\beta_i^P}{\delta^P} + \sigma \frac{\beta_i^I}{\mu + \alpha + \eta + \gamma^I} + (1 - \sigma) \frac{\beta_i^A}{\mu + \gamma^A} \right]$ because of differences in the local values of the transmission rates. Parameters and other details as in Fig. 1.

numbers, as shown in Supplementary Fig. 2. Finally, if $\mathcal{R}_0 > 1$ and $e_0 > 0$ (Fig. 1g–i), system trajectories exponentially diverge from the unstable DFE, thus giving rise to evident epidemic curves in terms of both disease prevalence and system output.

To integrate the examples shown in Fig. 1, it is possible to perform a systematic classification of the different types of disease dynamics produced by the two-community SEPIAR for different values of the local reproduction numbers $R_{01}$ and $R_{02}$, i.e., the values of the reproduction number in the two local communities evaluated as if neither outbound nor inbound mobility were allowed (Fig. 2). Local reproduction numbers can be obtained from the general expression (Eq. (1)), which accounts for human mobility) considering all connections as local (i.e., replacing the spatially explicit contact matrices with identity matrices). The shaded regions correspond to: (i) endemic establishment (red, $\mathcal{R}_0 > 1$, which by construction gives $e_0 > 0$); (ii) transient subthreshold epidemic (gray, $\mathcal{R}_0 < 1$, $e_0 > 0$); and (iii) rapid waning of the epidemic (blue, $e_0 < 0$, which normally also implies $\mathcal{R}_0 < 1$, save for nongeneric output transformations). As a reference, we also plot (black solid line) the parameter combinations where the population-weighted average $\tilde{\mathcal{R}}_0$ of the local reproduction numbers equals unity ($\tilde{\mathcal{R}}_0 = 1$). Local reproduction numbers appear to be only loosely related to the dynamics of the spatially explicit system. In regions ① and ② of the parameter space in Fig. 2, both local reproduction numbers are below one; however, while no epidemic waves are expected for the coupled spatial system in ①, the opposite is true in ②, where small outbreaks may in fact occur. By contrast, in region ③, one local reproduction number exceeds unity, yet neither endemicity nor outbreaks are expected for the connected system as a whole. In regions ④ and ⑤, one local reproduction number is above one —with a population-weighted average value $\tilde{\mathcal{R}}_0$ either below one (region ④) or above one (region ⑤). Outbreaks are expected, but no endemicity. In regions ⑥ and ⑦, one of the two local reproduction numbers is below one—with either $\tilde{\mathcal{R}}_0 < 1$ in ⑥ or $>1$ in ⑦—but large outbreaks and endemic persistence are possible in the spatial system. Finally, in region ⑧, both local reproduction numbers are above one, and large outbreaks are indeed possible in the spatially explicit model. Evidently, conclusions based on local reproduction numbers (or their average) would

**Table 2 Parameters and controls of model (Eq. (3)).**

| Parameter | Definition | Units | Value |
|---|---|---|---|
| $\mu$ | Baseline mortality rate | days$^{-1}$ | $3.65 \times 10^{-5}$ |
| $\beta_i^P$ | Transmission rate from post-latent individuals | days$^{-1}$ | $9.38 \times 10^{-1}$ |
| $\beta_i^I$ | Transmission rate from symptomatic individuals | days$^{-1}$ | $2.06 \times 10^{-2}$ |
| $\beta_i^A$ | Transmission rate from asymptomatic or paucisymptomatic individuals | days$^{-1}$ | $2.06 \times 10^{-2}$ |
| $\delta^E$ | Exit rate from the exposed class | days$^{-1}$ | $2.17 \times 10^{-1}$ |
| $\delta^P$ | Exit rate from the post-latent class | days$^{-1}$ | $5.00 \times 10^{-1}$ |
| $\sigma$ | Fraction of symptomatic infections | – | 0.25 |
| $\alpha$ | Disease-associated extra-mortality rate for symptomatic infections | days$^{-1}$ | $4.00 \times 10^{-2}$ |
| $\eta$ | Hospitalization rate of symptomatic individuals | days$^{-1}$ | $2.00 \times 10^{-1}$ |
| $\gamma^I$ | Recovery rate for symptomatic infections | days$^{-1}$ | $7.14 \times 10^{-2}$ |
| $\gamma^A$ | Recovery rate for asymptomatic or paucisymptomatic infections | days$^{-1}$ | $1.43 \times 10^{-1}$ |
| $r^X$ | Fraction of mobility-associated contacts for class $X \in \{S, E, P, A, R\}$ | – | $5.00 \times 10^{-1}$ |
| $r^I$ | Fraction of mobility-associated contacts for class $I$ | – | 0 |

| Control | Definition | Units | Value |
|---|---|---|---|
| $\epsilon_i$ | Contact rate reduction in community $i$ | – | [0, 1] |
| $\xi_{ij}$ | Travel restriction between communities $i$ and $j$ | – | [0, 1] |
| $\chi_i^X$ | Isolation rate for infected class $X \in \{E, P, I, A\}$ in community $i$ | days$^{-1}$ | $\geq 0$ |

Parameter values (median estimates at the beginning of the epidemic, when no controls were assumed to be in place) are taken from Bertuzzo et al.[14]. Control values are left free to vary within suitable ranges.

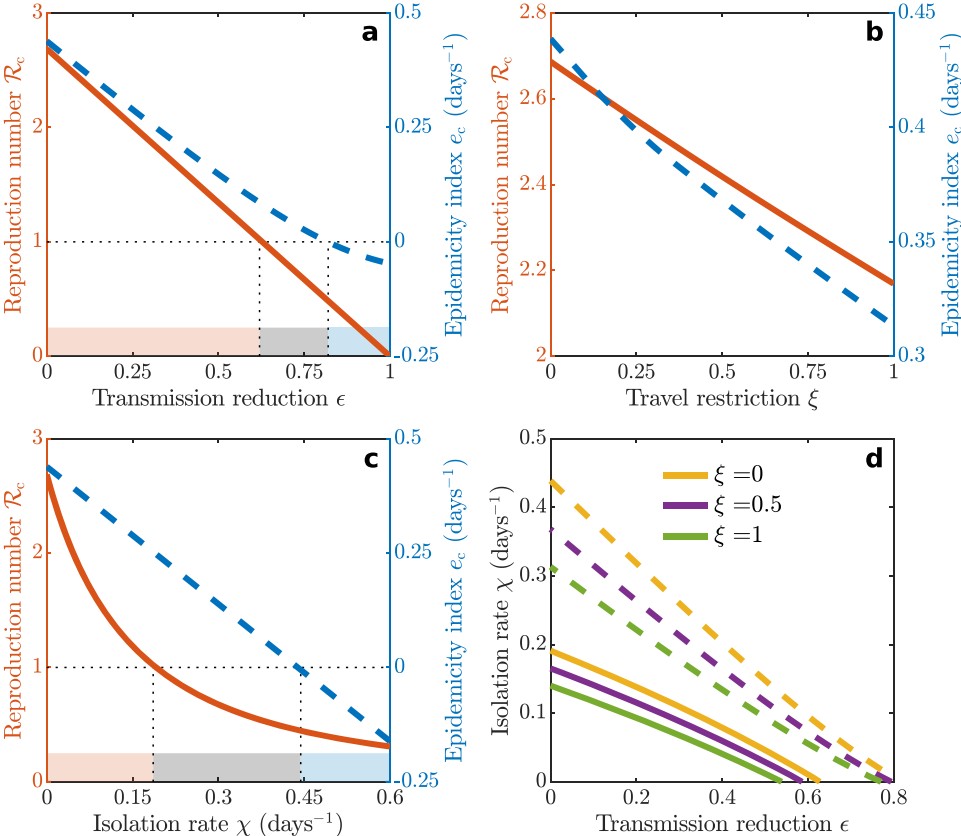

**Fig. 3 The effects of spatially homogeneous control measures on long-term endemicity and short-term epidemicity of COVID-19 in Italy. a** Plot of the relation between $\mathcal{R}_c$ (left axis, red), $e_c$ (right axis, blue), and the spatially uniform transmission rate reduction ($\epsilon_i = \epsilon$ for all $i$'s). **b** Same as (**a**) for the effects of travel restrictions ($\xi_{ij} = \xi$ for all $i$'s and $j$'s). **c** Same as (**a**) for the effects of the isolation of infected individuals ($\chi_i^X = \chi$ for all $i$'s and $X \in \{E, P, I, A\}$). **d** Simultaneous deployment of controls leading to $\mathcal{R}_c = 1$ (solid curves) or $e_c = 0$ (dashed) for three values of $\xi$, the imposed travel restriction. The DFE is asymptotically unstable ($\mathcal{R}_c > 1$) for parameter values below the solid curves and endowed with negative epidemicity ($e_c < 0$) above the dashed curves. In all panels, $w^X = 1$ ($X \in \{E, P, I, A\}$, see "Methods"). Colors at the bottom of **a**, **c** specify the ranges of endemicity/epidemicity conditions (red: $\mathcal{R}_c > 1$; gray: $\mathcal{R}_c < 1$, $e_c > 0$; blue: $\mathcal{R}_c < 1$, $e_c < 0$). Parameters as in Table 2.

often provide wrong indications about not only short-term epidemic dynamics, but also long-term disease-transmission patterns. The behavior of the two-community system for different configurations of population distribution and human mobility is shown in Supplementary Fig. 3.

**Effect of various control strategies in Italy**. We apply our methodology to the Italian COVID-19 epidemic ("Methods"). Model parameters are shown in Table 2 and were estimated in a Bayesian framework[14] for the early phase of the epidemic, when the disease was spreading largely unnoticed and no containment measures were yet in place ($\epsilon_i = 0$, $\chi_i^X = 0$, and $\xi_{ij} = 0$ for all $i$'s, $j$'s, and $X \in \{E, P, I, A\}$). Transmission rates, $\beta_i$, were reasonably[14] assumed to be spatially homogeneous over the whole country at the beginning of the epidemic ($\beta_i = \beta$ for all $i$'s). We also assume no pre-existing immunity within all communities ($S_i(0) = N_i$ for every node $i$). The estimated value of the basic reproduction number is $\mathcal{R}_0 = 2.7$, while the epidemicity index is $e_0 = 0.44$ day$^{-1}$. These figures are in line with the rapid spread of the COVID-19 epidemic in large portions of Italy after the first identified foci of infection[7]. Relying on its own database, Istituto Superiore di Sanità (ISS) released estimates of local reproduction numbers[38]. Thus, we also provided crude estimates of spatially heterogeneous transmission rates (Supplementary Table 1).

The model allows us to analyze the effects of containment strategies that might have prevented the occurrence of endemicity, based on the estimated threshold quantity $\mathcal{R}_c$. Key results are given

in Fig. 3. Therein, a significant reduction of the transmission rate ($\epsilon_i = \epsilon = 0.60$ for all $i$'s) is required to prevent endemic transmission if social distancing, use of personal protective equipment, and local mobility restriction are the sole control measures in action (a). Conversely, uniform travel restrictions ($\xi_{ij} = \xi$ for all $i$'s and $j$'s) can progressively reduce $\mathcal{R}_c$, but alone do not suffice to control it below one (b). As for the isolation of infected individuals, a spatially uniform removal rate ($\chi_i^X \equiv \chi$ for all $i$'s and $X \in \{E, P, I, A\}$) of ≈0.20 days$^{-1}$, corresponding to the daily isolation of ≈18% (i.e., $1 - e^{-0.20}$) of all infected individuals, is needed to achieve $\mathcal{R}_c < 1$ if no other measures are simultaneously enforced (c). When applied together, different controls act synergistically toward long-term endemicity suppression, as suggested earlier[14]. As an example, the combination of a 40% transmission reduction with an isolation rate of 0.05 days$^{-1}$ turns out to be as effective as a 20% transmission reduction combined with an isolation rate of 0.11 days$^{-1}$, provided that these measures are coupled with a 50% travel restriction (d).

The same measures used for long-term control can be applied also to preventively curtail short-term epidemic outbreaks that might arise if $e_c$ were larger than zero. If individually deployed, strong (>80%) transmission reductions (Fig. 3a) could theoretically achieve $e_c < 0$, differently from travel restrictions (b). Isolation of infected individuals (c) might also succeed in preventing positive epidemicity, although the whole sequence of actions (from testing of suspected cases to removal from the community) must occur rapidly (i.e., at a rate of 0.45 days$^{-1}$,

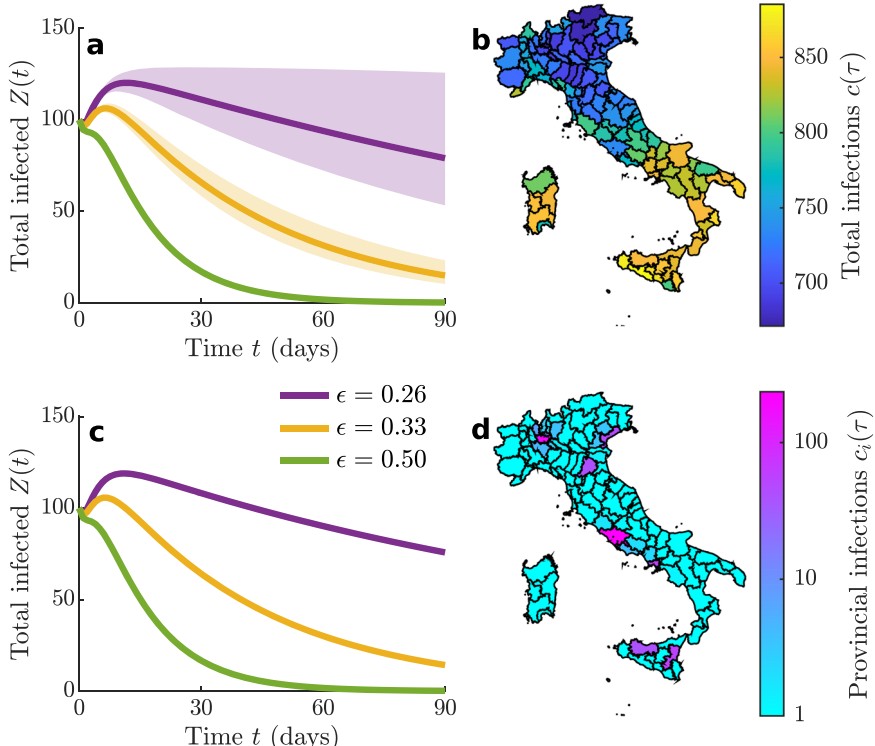

**Fig. 4 The effect of initial conditions and of containment measures on subthreshold epidemics in Italy.** The SEPIAR model has been numerically integrated for a timespan $\tau = 90$ days starting from different initial conditions, while assuming that spatially homogeneous containment measures are in place from the beginning of the epidemic. **a** Total number of infected individuals in the community, evaluated as $Z(t) = \sum_{i=1}^{n} E_i(t) + P_i(t) + I_i(t) + A_i(t)$, for outbreaks started by seeding one by one each of the 107 Italian provinces (solid lines: across-province median; shadings: min-max envelope) with an initial number of exposed individuals $E_i(0) = 100$, assuming an otherwise fully susceptible population ($S_i(0) = N_i - E_i(0)$). **b** Provinces in the map are color-coded according to the cumulated number of cases over the whole national territory up to the end of the simulation timespan for a subthreshold epidemic seeded in the considered province and for the intermediate-control scenario (yellow) of (**a**). **c** Same as (**a**) for a simulated outbreak obtained by seeding the provinces where the ten busiest Italian airports are located ("Methods"). A total number of 100 exposed individuals has been allocated proportionally to the total passenger flux at the beginning of the simulation. **d** Map of the projected infections in each province up to the end of the simulation period for the intermediate-control scenario (yellow) of (**c**). Parameters as in Table 2, with $\epsilon_i = \epsilon$ (numerical values are given in **c**), $\xi_{ij} = 0.5$, and $\chi_i^X = 0.1$ days$^{-1}$ for all $i$'s, $j$'s, and $X \in \{E, P, I, A\}$. For the three combinations of the control parameters shown in **a** and **c**, we find $\mathcal{R}_c \approx 0.99$, 0.90, and 0.67 for $\epsilon = 0.26$, 0.33, and 0.50, respectively, with $e_c$ (evaluated for $w^X = 1$, $X \in \{E, P, I, A\}$) $\approx 0.13$, 0.099, and 0.017 day$^{-1}$.

which corresponds to isolating about 36% of the total infected cases in one single day). Simultaneous deployment of control measures (d) thus seems like a promising pathway also toward the suppression of short-term epidemicity. As an example, a 40% transmission reduction combined with an isolation rate of 0.15 days$^{-1}$ (namely isolating 14% of the cases in one day) and 50% travel restrictions would prove effective in bringing the short-term epidemicity index below zero.

Quantitatively different results (Supplementary Fig. 4) are obtained if spatially heterogeneous transmission rates[38] are used (Supplementary Table 1). In particular, travel restrictions are more effective at reducing both $\mathcal{R}_c$ and $e_c$, yet still insufficient to prevent the occurrence of endemic transmission and positive epidemicity, if enforced alone. Heterogeneous transmission can be harnessed to prioritize the spatial deployment of preventive containment strategies, thereby highlighting trade-offs between dispersing containment efforts over large areas vs. focusing them on smaller ones (Supplementary Fig. 5). This analysis suggests that reduction of inter-individual transmission, e.g., by using personal protective equipment, enforcing social distancing, and limiting local-scale mobility, is best applied at large (e.g., country-wide) scales, while mass-testing and isolation of infected individuals can also be effective if enforced within the most-at-risk areas (e.g., province scale).

**Subthreshold epidemic containment**. A subthreshold epidemic during containment phases is a surge in the number of infections generated by seeding new cases when $\mathcal{R}_c < 1$ and $e_c > 0$ ("Methods"). Flare-ups simulated by SEPIAR typically exhibit unimodal shapes similar to those shown in Fig. 1e, f. Whether or not these epidemic waves prove dangerous depends on the location and size of the initial hotbed, and on the chances of possible coalescence with neighboring foci. For the Italian case, Fig. 4 shows the effect of the location of the initial seeding (100 exposed individuals) and containment measures (subsumed by three different values of $\epsilon$, coupled with assigned isolation rate and travel restriction) on subthreshold epidemic dynamics. Depending on the strength of the containment measures, outbreaks where only one province is initially affected may either grow considerably over time and last long or wane rapidly without resulting in a large number of country-wide total infections (Fig. 4a). The cumulated number of cases, evaluated over a fixed time span of three months, depends on the seeding location (b), and is affected by both the total magnitude and the duration of an outbreak (in this respect, Fig. 4b represents an underestimation of the total case counts associated with the outbreaks, as many of them may still be ongoing at the end of the simulation period). The epidemic trajectories obtained when the outbreak starts from the busiest Italian airports ("Methods"), e.g., via incoming infected

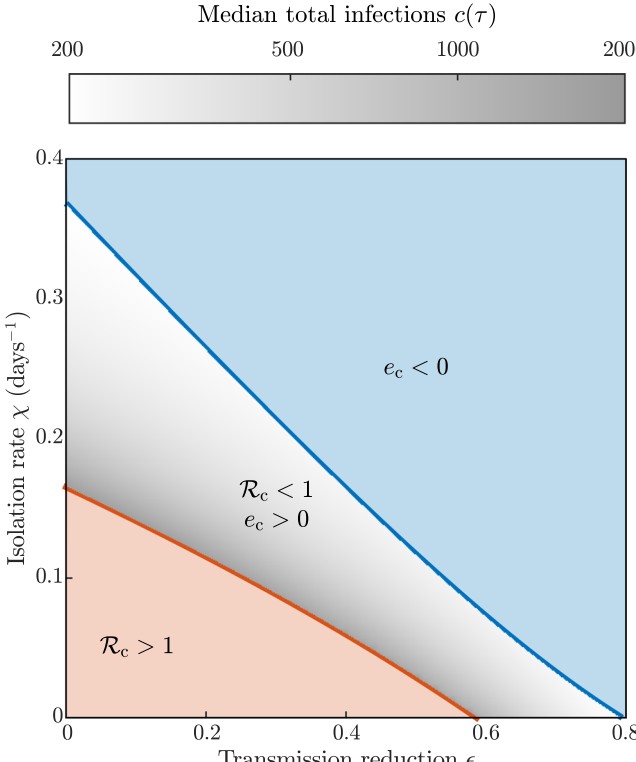

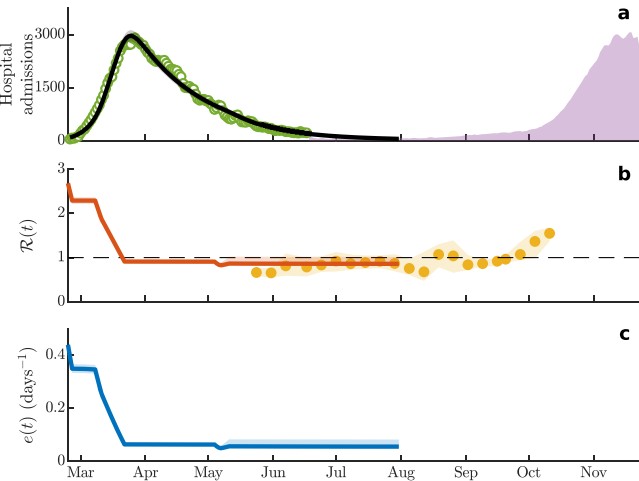

**Fig. 6 Effective reproduction number $\mathcal{R}(t)$ and epidemicity index $e(t)$ for the first wave of the COVID-19 pandemic in Italy. a** Hospital admissions (March to November 2020). Green empty dots represent the curated data that has been used[14] to calibrate the SEPIAR model (the black line and black shading are the median and the 95% confidence interval, respectively, of 2000 simulations with parameter values drawn from a posterior distribution estimated from data). The SEPIAR model has been run until the end of July for validation. Purple shading shows data not used for calibration. **b** Temporal dynamics of the effective reproduction number obtained from SEPIAR (red curve: median; red shading: 95% confidence interval). Also shown (yellow dots) is the timeseries of effective reproduction numbers and their confidence interval (yellow shading) as published by ISS[51]. **c** Effective epidemicity index (blue curve: median; blue shading: 95% confidence interval) computed from SEPIAR ("Methods").

**Fig. 5 The effect of spatially homogeneous controls on subthreshold epidemic size.** The SEPIAR model has been simulated seeding one by one each of the 107 Italian provinces with an initial abundance of exposed individuals $E_i(0) = 100$ (details as in Fig. 4a), for parameter combinations resulting in $\mathcal{R}_c < 1$ (above the red solid curve) and $e_c > 0$ (below the blue dashed curve). Gray shading represents the total number of cases over the whole national territory $c(\tau)$ up to the end of the simulation timespan ($\tau = 90$ days), evaluated as the median of the values obtained with different initial conditions (i.e., outbreaks starting from each of the different provinces). Parameters as in Table 2, with $\epsilon_i = \epsilon$, $\xi_{ij} = 0.5$, $\chi_i^X = \chi$ for all $i$'s, $j$'s, and $X \in \{E, P, I, A\}$, and $w^X = 1$ ($X \in \{E, P, I, A\}$).

passengers, are shown in Fig. 4c, and may also result in a significant number of cases with respect to the initial seeding. In this case, infections tend to concentrate in the provinces closest to the airports (d), owing to mobility matrices that largely reflect daily commuting.

The roles of spatially homogeneous transmission reduction ($\epsilon$) and isolation rate ($\chi$) on the size of subthreshold epidemics are considered in detail in Fig. 5 for given (halved) mobility (travel restrictions prove less effective than other measures, Fig. 3d). Gray scale codes the country-wide number of cases 90 days after the beginning of the outbreak, evaluated as the median of the values obtained for all subthreshold epidemics starting from single-province seeding (i.e., the same initial condition as in Fig. 4a, b). The closer the combination of parameters brings to the frontier $\mathcal{R}_c = 1$, the steeper is the increase in total infections (up to around 2000 cases generated by the initial seeding of 100 exposed individuals—which again represents a lower-bound estimate for outbreaks that are not yet over three months after they started), reflecting the transcritical bifurcation of the DFE. Cost–benefit analysis based on combinations of interventions should always include the differential costs of isolation rates versus transmission reductions. Benefits of isolation would be negligible for transmission reductions beyond 80%. The same would occur for transmission reductions, should isolation rates be larger than 0.38 days$^{-1}$.

**Effective reproduction number and epidemicity index.** The spatially explicit structure of SEPIAR allows us also to analyze containment strategies aimed at breaking transmission if deployed once the epidemic has already started. To that end, numerical simulations can be usefully complemented by the evaluation through time of the effective reproduction number, $\mathcal{R}(t)$, and of the effective epidemicity index, $e(t)$ ("Methods"). These two indices are computed by updating the fraction of susceptible individuals and the epidemiological parameters at time $t$ in the general expressions for $\mathcal{R}_c$ and $e_c$. Supplementary Figures 6–9 indicate that both contact reduction and isolation of infected individuals can significantly reduce disease transmission, with a clear spatial gradient whereby country-scale interventions are the most effective while province-scale interventions are the least effective. Indeed, only a strong, large-scale deployment of containment measures can bring $\mathcal{R}(t)$ below one and, possibly, $e(t)$ below zero.

Finally, Fig. 6 shows the results of the computation of the effective epidemicity index for the first wave of the COVID-19 pandemic in Italy, which peaked at the end of March 2020 (a). A time-varying parameterization of SEPIAR[14], accounting for the progressive tightening of containment measures and reduction of human mobility ("Methods"), shows how $\mathcal{R}(t)$ and $e(t)$ varied over time—mostly as a response to the change in the transmission parameters induced by the application of control measures, echoing modeling results concerning the effectiveness of control measures on early transmission dynamics[7,39]. By contrast, only marginally did the dynamics of the susceptible compartment affect $\mathcal{R}(t)$ and $e(t)$, consistently with model-based estimates suggesting that the depletion of the susceptible pool during the first wave of the pandemic has been small to negligible in northern Italy, and basically null in the south of the country[14]. The model suggests that $\mathcal{R}(t)$ has been below the threshold value

of one for several months, starting a few weeks after the country went into full lockdown and well into the summer (b). These modeling results agree well with data-based estimates of the effective reproduction number made available by ISS, which confirm that $\mathcal{R}(t)$ has been below the critical threshold (or close to it) until the end of September 2020. However, during August to September, cases were already on a rise that culminated with a peak in late November 2020 (see again a). On the other hand, the effective epidemicity index always remained positive despite all containment efforts, signaling a continuing risk of subthreshold epidemic recurrence.

## Discussion

Most emerging infectious diseases are zoonoses[40], suggesting overarching themes connecting ecology to epidemiology. The theory of epidemicity, which we have developed here for COVID-19 metapopulation models[7,14], is elemental to linking the reactivity of transient species dynamics in ecology[24,25] with the control of human infectious diseases, a valuable ecosystem service arising from biodiversity conservation[41]. The definition of epidemicity indices specifically allows to establish necessary conditions for the occurrence of subthreshold outbreaks, i.e., for the development of epidemic transmission when the reproduction number is below unity.

Some affinities exist between epidemicity analysis and the literature on so-called stuttering transmission chains[42], which are typical of pathogens spreading inefficiently in a population. In that case, however, the estimation of quantities like the total size of an outbreak is typically done under the hypothesis that the average number of cases (evaluated over different observations of the process) declines monotonically over time when $\mathcal{R}_0 < 1$[43,44]. Thus, the possibility of subthreshold yet non-negligible outbreaks is left unexplored. The same observation holds as well with respect to the vast body of literature devoted to the study of epidemic dynamics in stochastic settings, which is mostly focused on the case $\mathcal{R}_0 \geq 1$, under the assumption that an epidemic would shrink exponentially otherwise[45–50]. By contrast, containment strategies aimed at breaking the transmission chain of subthreshold epidemics find in our theory an objective tool to rank the efficacy of collective responses to recurrent spatial foci of infection that may synchronize into large-scale infection outbreaks. However, open issues remain toward a general characterization of epidemicity. One is the operational definition of the geographical boundaries of the community where scenarios of disease spread can be made. Multi-scale approaches may help to that end. They are of routine use in other branches of science—like limited area models in meteorology, climatology, and oceanography, which use larger-scale models to impose boundary conditions. In a pandemic context, this approach may be particularly useful to assess international travel as a means of infection propagation.

Concerning the application to the Italian case study, Fig. 6 conveys an important message, because it provides an example of a large-scale epidemic where the effective reproduction number, $\mathcal{R}(t)$, was below one for a sizable portion of the epidemiological trajectory (at least four months), while the effective epidemicity index, $e(t)$ remained positive. The agreement between the subthreshold values of $\mathcal{R}(t)$ estimated from either modeling (with SEPIAR) or data (by ISS, the Italian agency in charge of COVID-19 epidemiology[51]) is substantial and, given the broad differences in the methodological assumptions, noteworthy per se. Positive values of $e(t)$, evaluated by means of SEPIAR and the methodology proposed here, indicated the extant danger of flare-ups allowing subthreshold circulation of the virus. From a disease-control perspective, the persistence of positive values of the epidemicity index should have prompted a preventive employment of stricter control measures during the summer. All these

considerations suggest that the reproduction number, a fundamental long-term diagnostic indicator, may actually bear little prognostic power when its value is below the critical threshold and should be complemented with the evaluation of the epidemicity index, which reveals to be key in detecting the short-term reactivity of the core disease-transmission system. Had $e(t) < 0$ been achieved (implying $\mathcal{R}_t < 1$), even short-term, subthreshold epidemics would have been prevented, thus curtailing the spatial transmission of the pathogen—with possible implications for the second, nation-wide epidemic wave that Italy experienced in the fall.

The temporal dynamics of the effective reproduction number and epidemicity index suggest that a continuous monitoring of these quantities may be crucial to capture the effects of containment measures. In this respect, a promising linkage between ex-post, calibration-based vs. data-based estimates could be represented by data assimilation schemes (e.g., Ensemble Kalman Filter), in which state and model parameters are jointly and repeatedly re-evaluated over time. These techniques have been recently used to analyze cholera transmission in Haiti[52] in conjunction with spatially explicit metapopulation models not unlike SEPIAR (albeit representing different underlying epidemiological dynamics). Joint, frequent updates of parameter values and state variables may allow, on one hand, the reliable evaluation of the effective reproduction number and the epidemicity index, as well as their temporal (and possibly seasonal) dynamics; and, on the other, the production of reasonable epidemiological projections with a lead time of at least a few weeks. An integrative approach like the one just outlined here would clearly represent an important extension to our modeling framework.

Growing pandemic figures require reliable assessments of real-time control measures, and the epidemicity index is suggested to be a useful addition to the tools currently shaping emergency management policies—specifically, as a synthetic measure that, based on the epidemiological parameters and their variations, may signal the risk of possible subthreshold epidemics. The examples presented here illustrate how the epidemicity index can be used to complement and specialize the variety of existing reproduction numbers by highlighting control measures that are effective for both short- and long-term controls. We thus conclude that the epidemicity index should find many applications in the epidemiology of emerging infectious diseases, and should be included in cost–benefit analysis of alternative intervention options.

## Methods

**Data and data processing**. The modeling tools described in the following sections are applied to the Italian COVID-19 epidemic at the scale of second-level administrative divisions, i.e., provinces and metropolitan cities (as of 2020, 107 spatial units). Official data about resident population at the provincial level are produced yearly by the Italian National Institute of Statistics (Istituto Nazionale di Statistica, ISTAT; data available at http://dati.istat.it/Index.aspx?QueryId=18460). The January 2019 update has been used to inform the spatial distribution of the population.

The data to quantify nation-wide human mobility prior to the pandemic come from ISTAT (specifically, from the 2011 national census; data available online at https://www.istat.it/it/archivio/139381). Mobility fluxes, mostly reflecting commuting patterns related to work and study purposes, are provided at the scale of third-level administrative units (municipalities)[53,54]. These fluxes were upscaled to the provincial level following the administrative divisions of 2019, and used to evaluate the fraction $p_i$ of mobile people and the fraction $q_{ij}$ of mobile people between $i$ and all other administrative units $j$ (see Supplementary Material in Gatto et al.[7]).

Airport traffic data for year 2019, used to inform the simulation shown in Fig. 4c, d, are from the Italian Airports Association (Assaeroporti; data available at http://assaeroporti.com/statistiche_201912/). Note that airports have been assigned to the main Metropolitan Area they serve, rather than to the province where they are geographically located (e.g., Malpensa Airport has been assigned to the Metropolitan City of Milano, rather than to the neighboring Varese province, where it actually lies).

Model parameters are taken from a paper by Bertuzzo et al.[14], where they were inferred in a Bayesian framework on the basis of the official epidemiological bulletins released daily by Dipartimento della Protezione Civile[55] (data available online at https://github.com/pcm-dpc/COVID-19) and the bulletins of Epicentro, at ISS[51,56]. The parameters estimated for the initial phase of the Italian COVID-19 epidemic[14], during which SARS-CoV-2 was spreading unnoticed in the population, reflect a situation of unperturbed social mixing and human mobility, absent any effort devoted to disease control. This parameterization, in which all parameters (including the transmission rates) are spatially homogeneous, is reported in Table 2 and has been used to produce all the results presented in the main text, except for those of Fig. 6. In this case, to account for the containment measures put in place by the Italian authorities and their effects on transmission rates and mobility patterns during the first months of the pandemic, a time-varying parameterization[14] for the period February 24 to May 1, 2020 has been used. In this parameterization, the transmission rates were allowed to take different values over different time windows, corresponding to the timing of the implementation of the main nation-wide restrictions, or lifting thereof. Specifically, the effect of the containment measures was parameterized by assuming that the transmission parameters had a sharp decrease after the containment measures announced at the end of February and the beginning of March, and that they were further reduced in the following weeks as the country was effectively entering full lockdown. As a by-product, these time-varying transmission rates can also at least partially account for seasonal effects on disease transmission. Due to the emerging nature of the pathogen, seasonality has not been given further consideration in this work; however, it may become a key component of future modeling efforts aimed at studying post-pandemic SARS-CoV-2 transmission dynamics[3], i.e., if/when the pathogen establishes as endemic. Spatial connectivity too was modified with respect to the baseline scenario to reflect the disruption of mobility patterns induced by the pandemic and the associated containment measures[14]. Specifically, between-province mobility was progressively reduced as the epidemic unfolded according to estimates obtained through mobility data from mobile applications[53,57].

**Spatially explicit SEPIAR with distributed controls**. We consider a set of $n$ communities connected by human mobility fluxes. In each community, the human population is subdivided according to infection status into the epidemiological compartments of susceptible, exposed (latently infected), post-latent (incubating infectious, also termed pre-symptomatic[7]), symptomatic infectious, asymptomatic infectious (including paucisymptomatic), and recovered individuals. The present model utilizes previous work aimed to describe the first wave of COVID-19 infections[7,14]. In particular, it allows us to account for three widely adopted types of containment measures: reduction of local transmission (as a result of the use of personal protections, social distancing, and local mobility restriction), travel restriction, and isolation of infected individuals. To describe the effects of isolation, each infected compartment (exposed, post-latent, symptomatic and asymptomatic) is actually split into two, which allows keeping track of the abundances of infected individuals who are still in the community vs. those who are removed from it (i.e., either in isolation at a hospital, if symptomatic, or quarantined at home, if exposed, post-latent, or asymptomatic). The state variables of the model are summarized in Table 1. Supplementary Figure 1 recapitulates the structure of the model.

COVID-19 transmission dynamics are thus described by the following set of ordinary differential equations:

$$
\begin{aligned}
\dot{S}_i &= \mu(N_i - S_i) - \lambda_i S_i \\
\dot{E}_i &= \lambda_i S_i - (\mu + \delta^E + \chi_i^E)E_i \\
\dot{P}_i &= \delta^E E_i - (\mu + \delta^P + \chi_i^P)P_i \\
\dot{I}_i &= \sigma\delta^P P_i - (\mu + \alpha + \gamma^I + \eta + \chi_i^I)I_i \\
\dot{A}_i &= (1-\sigma)\delta^P P_i - (\mu + \gamma^A + \chi_i^A)A_i \\
\dot{E}_i^q &= \chi_i^E E_i - (\mu + \delta^E)E_i^q \\
\dot{P}_i^q &= \chi_i^P P_i + \delta^E E_i^q - (\mu + \delta^P)P_i^q \\
\dot{I}_i^h &= (\eta + \chi_i^I)I_i + \sigma\delta^P P_i^q - (\mu + \alpha + \gamma^I)I_i^h \\
\dot{A}_i^q &= \chi_i^A A_i + (1-\sigma)\delta^P P_i^q - (\mu + \gamma^A)A_i^q \\
\dot{R}_i &= \gamma^I(I_i + I_i^h) + \gamma^A(A_i + A_i^q) - \mu R_i.
\end{aligned}
\tag{3}
$$

Susceptible individuals are recruited into community $i$ ($i = 1\ldots n$) at a constant rate $\mu N_i$, with $\mu$ and $N_i$ being the average mortality rate of the population and the size of the community in the absence of disease, respectively, and die at rate $\mu$. In this way, the equilibrium size of community $i$ without disease amounts to $N_i$. Susceptible individuals get exposed to the pathogen at rate $\lambda_i$, corresponding to the force of infection for community $i$ (detailed below), thus becoming latently infected (but not infectious yet). Exposed individuals die at rate $\mu$ and transition to the post-latent, infectious stage at rate $\delta^E$. If containment measures including mass testing and preventive isolation of positive cases are in place, exposed individuals may be removed from the general population and quarantined at rate $\chi_i^E$. Post-latent individuals die at rate $\mu$, progress to the next infectious classes at rate $\eta^P$, developing an infection that can be either symptomatic—with probability $\sigma$—or asymptomatic, including the case in which only mild symptoms are present—with probability $1 - \sigma$, and may be tested and quarantined at rate $\chi_i^P$. Symptomatic

infectious individuals die at rate $\mu + \alpha$, with $\alpha$ being an extra-mortality term associated with disease-related complications, recover from infection at rate $\gamma^I$, may spontaneously seek treatment at a hospital at rate $\eta$, and may be identified through mass screening and hospitalized at rate $\chi_i^I$. Asymptomatic individuals die at rate $\mu$, recover at rate $\gamma^A$, and may be quarantined at rate $\chi_i^A$. Infected individuals who are either hospitalized or quarantined at home are subject to the same epidemiological dynamics as those who are still in the community, but are considered to be effectively removed from it, thus not contributing to disease transmission. Individuals who recover from the infection die at rate $\mu$, and are assumed to have permanent immunity to reinfection. This last assumption is not fundamental, as loss of immunity can be easily included in the model. However, immunity to SARS-CoV-2 reinfection is reported to be relatively long-lasting (a few months at least), hence its loss cannot alter transmission dynamics over epidemic timescales[14].

The cornerstone of model (Eq. (3)) is the force of infection, $\lambda_i$, which in a spatially explicit setting must account not only for locally acquired infections but also for the role played by human mobility. We assume that, at the spatiotemporal scales of interest for our problem, human mobility mostly depicts daily commuting flows (also coherently with the data available for parameterization; see above) and does not actually entail a permanent relocation of individuals. We thus describe human mobility (and the associated social contacts possibly conducive to disease transmission) by means of instantaneous spatial-mixing matrices $M_{c,ij}^X$ (with $X \in \{S, E, P, I, A, R\}$), i.e.,

$$
M_{c,ij}^X = \begin{cases} r^X p_i q_{ij}(1 - \xi_{ij}) & \text{if } i \neq j \\ (1 - p_i) + (1 - r^X)p_i + r^X p_i q_{ij}(1 - \xi_{ij}) & \text{if } i = j, \end{cases}
\tag{4}
$$

where $p_i$ ($0 \leq p_i \leq 1$ for all $i$'s) is the fraction of mobile people in community $i$, $q_{ij}$ ($0 \leq q_{ij} \leq 1$ for all $i$'s and $j$'s) represents the fraction of people moving between $i$ and $j$ (including $j = i$, $\sum_{j=1}^n q_{ij} = 1$ for all $i$'s), $r^X$ ($0 \leq r^X \leq 1$ for all $X$'s) quantifies the fraction of contacts occurring while individuals in epidemiological compartment $X$ are traveling, and $\xi_{ij}$ ($0 \leq \xi_{ij} \leq 1$ for all $i$'s and $j$'s) represents the effects of travel restrictions that may be imposed between any two communities $i$ and $j$ as a part of the containment response. Therefore, the probability that residents from $i$ have social contacts while being in $j$ (independently of with whom) is assumed to be proportional to the fraction $r^X$ of the mobility-related contacts of the individuals in epidemiological compartment $X$, multiplied by the probability $p_i$ that people from $i$ travel (independently of the destination) and the probability $q_{ij}$ that the travel occurs between $i$ and $j$, possibly reduced by a factor $1 - \xi_{ij}$ accounting for travel restrictions. All other contacts contribute to mixing within the local community ($i$ in this case). Note also that if $\xi_{ij} = 0$ for all $i$'s and $j$'s, then $M_{c,ij}^X$ reduces to $M_{ij}^X$, i.e., to the mixing matrix in the absence of disease-containment measures. In this case, $\sum_{j=1}^n M_{ij}^X = 1$ for all $i$'s and $X$'s. It is important to remark, though, that the epidemiologically relevant contacts between the residents of two different communities, say $i$ and $j$, may not necessarily occur in either $i$ or $j$; in fact, they could happen anywhere else, say in community $k$, between residents of $i$ and $j$ simultaneously traveling to $k$. On this basis, we define the force of infection as

$$
\lambda_i = \sum_{j=1}^n M_{c,ij}^S \frac{(1 - \epsilon_j)\left(\beta_j^P \sum_{k=1}^n M_{c,kj}^P P_k + \beta_j^I \sum_{k=1}^n M_{c,kj}^I I_k + \beta_j^A \sum_{k=1}^n M_{c,kj}^A A_k\right)}{\sum_{k=1}^n \left(M_{c,kj}^S S_k + M_{c,kj}^E E_k + M_{c,kj}^P P_k + M_{c,kj}^I I_k + M_{c,kj}^A A_k + M_{c,kj}^R R_k\right)},
\tag{5}
$$

where the parameters $\beta_j^X$ ($X \in \{P, I, A\}$) are the community-dependent rates of disease transmission from the three infectious classes, $\epsilon_j$ ($0 \leq \epsilon_j \leq 1$ for all $j$'s) represents the reduction of transmission induced by social distancing, the use of personal protective equipment, and local mobility restrictions if such containment measures are in fact in place, and the terms $M_{c,ij}^X$ (with $X \in \{S, E, P, I, A, R\}$) describe the epidemiological effects of mobility between $i$ and $j$ in the presence of disease-containment measures. Note that transmission has been assumed to be frequency-dependent.

The parameters $\mu$, $\delta^X$ ($X \in \{E, P\}$), $\sigma$, $\alpha$, $\eta$, $\gamma^X$ ($X \in \{I, A\}$), and $r^X$ ($X \in \{S, E, P, I, A, R\}$) are assumed to be community-independent, for they pertain to population demography at the country scale or the clinical course of the disease. By contrast, the transmission rates $\beta_i^X$ ($X \in \{P, I, A\}$) and the control parameters, namely the isolation rates $\chi_i^X$ ($X \in \{E, P, I, A\}$), the reductions of transmission due to personal protection, social distancing, and local mobility restriction $\epsilon_i$, and the travel restrictions $\xi_{ij}$, are assumed to be possibly community-dependent, thereby reflecting spatial heterogeneities in disease transmission prior to the implementation of containment measures ($\beta_i^X$), testing effort and/or strategy ($\chi_i^X$), local transmission reduction ($\epsilon_i$), and travel restriction ($\xi_{ij}$).

**Derivation of the basic and control reproduction numbers**. Close to the DFE, a state in which all individuals are susceptible to the disease ($S_i = N_i$, with $N_i$ being the baseline population size of community $i$) and all the other epidemiological compartments are empty ($E_i = P_i = I_i = A_i = E_i^q = P_i^q = I_i^h = A_i^q = R_i = 0$ for all $i$'s), the dynamics of model (Eq. (3)) is described by the linearized system $\dot{\mathbf{x}} = \mathbf{J_c}\mathbf{x}$, where $\mathbf{x} = [S_i, E_i, P_i, I_i, A_i, E_i^q, P_i^q, I_i^h, A_i^q, R_i]^T$ (where $i = 1\ldots n$ and the

superscript $T$ denotes matrix transposition) and $\mathbf{J_c}$ is the spatial Jacobian matrix

$$
\mathbf{J_c} =
\begin{bmatrix}
-\mu\mathbf{I} & \mathbf{0} & -\theta_c^P & -\theta_c^I & -\theta_c^A & \mathbf{0} & & \mathbf{0} & \mathbf{0} & \mathbf{0} \\
\mathbf{0} & -\phi_c^E & \theta_c^P & \theta_c^I & \theta_c^A & \mathbf{0} & & \mathbf{0} & \mathbf{0} & \mathbf{0} \\
\mathbf{0} & \delta^E\mathbf{I} & -\phi_c^P & \mathbf{0} & \mathbf{0} & \mathbf{0} & & \mathbf{0} & \mathbf{0} & \mathbf{0} \\
\mathbf{0} & \mathbf{0} & \sigma\delta^P\mathbf{I} & -\phi_c^I & \mathbf{0} & \mathbf{0} & & \mathbf{0} & \mathbf{0} & \mathbf{0} \\
\mathbf{0} & \mathbf{0} & (1-\sigma)\delta^P\mathbf{I} & \mathbf{0} & -\phi_c^A & \mathbf{0} & & \mathbf{0} & \mathbf{0} & \mathbf{0} \\
\mathbf{0} & \chi^E & \mathbf{0} & \mathbf{0} & \mathbf{0} & -(\mu+\delta^E)\mathbf{I} & & \mathbf{0} & \mathbf{0} & \mathbf{0} \\
\mathbf{0} & \mathbf{0} & \chi^P & \mathbf{0} & \mathbf{0} & \delta^E\mathbf{I} & -(\mu+\delta^P)\mathbf{I} & \mathbf{0} & \mathbf{0} & \mathbf{0} \\
\mathbf{0} & \mathbf{0} & \mathbf{0} & \eta\mathbf{I}+\chi^I & \mathbf{0} & \mathbf{0} & \sigma\delta^P\mathbf{I} & -(\mu+\alpha+\gamma^I)\mathbf{I} & \mathbf{0} & \mathbf{0} \\
\mathbf{0} & \mathbf{0} & \mathbf{0} & \mathbf{0} & \chi^A & \mathbf{0} & (1-\sigma)\delta^P\mathbf{I} & \mathbf{0} & -(\mu+\gamma^A)\mathbf{I} & \mathbf{0} \\
\mathbf{0} & \mathbf{0} & \mathbf{0} & \gamma^I\mathbf{I} & \gamma^A\mathbf{I} & \mathbf{0} & & \gamma^I\mathbf{I} & \gamma^A\mathbf{I} & -\mu\mathbf{I}
\end{bmatrix},
$$

(6)

where $\mathbf{I}$ and $\mathbf{0}$ are the identity and null matrices of size $n$, respectively, $\phi_c^X$ ($X \in \{E, P, I, A\}$) are diagonal matrices whose non-zero elements are $\mu + \delta^E + \chi_i^E$ (for $\phi_c^E$), $\mu + \delta^P + \chi_i^P$ (for $\phi_c^P$), $\mu + \alpha + \eta + \gamma^I + \chi_i^I$ (for $\phi_c^I$), and $\mu + \gamma^A + \chi_i^A$ (for $\phi_c^A$), and the matrices $\theta_c^X$ ($X \in \{P, I, A\}$) are given by

$$
\theta_c^X = \mathbf{N}\mathbf{M}_c^S(\mathbf{I}-\epsilon)\beta^X(\Delta_c)^{-1}(\mathbf{M}_c^X)^T,
$$

(7)

where $\mathbf{N}$ is a diagonal matrix whose non-zero elements are the population sizes $N_i$, $\mathbf{M}_c^X = [M_{c,ij}^X]$ ($X \in \{S, P, I, A\}$) are sub-stochastic matrices representing the spatially explicit contact terms in the presence of containment measures, $\epsilon$ is a diagonal matrix whose non-zero entries are the transmission reductions $\epsilon_i$, $\beta^X$ ($X \in \{P, I, A\}$) are diagonal matrices whose non-zero elements are the contact rates $\beta_i^X$, and $\Delta_c$ is a diagonal matrix whose non-zero entries are the elements of vector $\mathbf{u}\mathbf{N}\mathbf{M}_c^S$, with $\mathbf{u}$ being a unitary row vector of size $n$.

Because of its block-triangular structure, it is immediate to see that $\mathbf{J_c}$ has $6n$ strictly negative eigenvalues, namely $-\mu$, with multiplicity $2n$, and $-(\mu+\delta^E)$, $-(\mu+\delta^P)$, $-(\mu+\alpha+\gamma^I)$, and $-(\mu+\gamma^A)$, each with multiplicity $n$. Therefore, the asymptotic stability properties of the DFE of model (Eq. (3)), which determine whether long-term disease circulation in the presence of controls is possible, are linked to the eigenvalues of a reduced-order spatial Jacobian associated with the infection subsystem, i.e., the subset of state variables directly related to disease transmission, in this case $\{E_1, \ldots, E_n, P_1, \ldots, P_n, I_1, \ldots, I_n, A_1, \ldots, A_n\}$. Note that introducing waning immunity would not change the spectral properties of the Jacobian matrix evaluated at the DFE. The reduced-order Jacobian $\mathbf{J_c^*}$ thus reads

$$
\mathbf{J_c^*} =
\begin{bmatrix}
-\phi_c^E & \theta_c^P & \theta_c^I & \theta_c^A \\
\delta^E\mathbf{I} & -\phi_c^P & \mathbf{0} & \mathbf{0} \\
\mathbf{0} & \sigma\delta^P\mathbf{I} & -\phi_c^I & \mathbf{0} \\
\mathbf{0} & (1-\sigma)\delta^P\mathbf{I} & \mathbf{0} & -\phi_c^A
\end{bmatrix}.
$$

(8)

The asymptotic stability properties of the DFE can be assessed through a NGM approach[22,37]. In fact, the spectral radius of the NGM provides an estimate of the so-called control reproduction number[58], $\mathcal{R}_c$, which can be thought of as the average number of secondary infections produced by one infected individual in a completely susceptible population in the presence of disease-containment measures. Clearly, if $\mathcal{R}_c > 1$ the pathogen can invade the population in the long run, and endemic transmission will eventually be established despite the implementation of disease-containment measures. To evaluate $\mathcal{R}_c$ for model (Eq. (3)), the Jacobian of the infection subsystem can be decomposed into a spatial transmission matrix

$$
\mathbf{T_c} =
\begin{bmatrix}
\mathbf{0} & \theta_c^P & \theta_c^I & \theta_c^A \\
\mathbf{0} & \mathbf{0} & \mathbf{0} & \mathbf{0} \\
\mathbf{0} & \mathbf{0} & \mathbf{0} & \mathbf{0} \\
\mathbf{0} & \mathbf{0} & \mathbf{0} & \mathbf{0}
\end{bmatrix},
$$

(9)

and a transition matrix

$$
\Sigma_c =
\begin{bmatrix}
-\phi_c^E & \mathbf{0} & \mathbf{0} & \mathbf{0} \\
\delta^E\mathbf{I} & -\phi_c^P & \mathbf{0} & \mathbf{0} \\
\mathbf{0} & \sigma\delta^P\mathbf{I} & -\phi_c^I & \mathbf{0} \\
\mathbf{0} & (1-\sigma)\delta^P\mathbf{I} & \mathbf{0} & -\phi_c^A
\end{bmatrix},
$$

(10)

so that $\mathbf{J_c} = \mathbf{T_c} + \Sigma_c$. The spatial NGM with large domain $\mathbf{K_c^L}$, including variables other than the states-at-infection[59] (i.e., the exposed individuals $E_i$) thus reads

$$
\mathbf{K_c^L} = -\mathbf{T_c}(\Sigma_c)^{-1} =
\begin{bmatrix}
\mathbf{K_c^1} & \mathbf{K_c^2} & \mathbf{K_c^3} & \mathbf{K_c^4} \\
\mathbf{0} & \mathbf{0} & \mathbf{0} & \mathbf{0} \\
\mathbf{0} & \mathbf{0} & \mathbf{0} & \mathbf{0} \\
\mathbf{0} & \mathbf{0} & \mathbf{0} & \mathbf{0}
\end{bmatrix},
$$

(11)

with

$$
\mathbf{K_c^1} = \delta^E\left[\theta_c^P + \sigma\delta^P\theta_c^I(\phi_c^I)^{-1} + (1-\sigma)\delta^P\theta_c^A(\phi_c^A)^{-1}\right](\phi_c^E)^{-1}(\phi_c^P)^{-1}
$$

$$
\mathbf{K_c^2} = \left[\theta_c^P + \sigma\delta^P\theta_c^I(\phi_c^I)^{-1} + (1-\sigma)\delta^P\theta_c^A(\phi_c^A)^{-1}\right](\phi_c^P)^{-1}
$$

$$
\mathbf{K_c^3} = \theta_c^I(\phi_c^I)^{-1}
$$

$$
\mathbf{K_c^4} = \theta_c^A(\phi_c^A)^{-1}.
$$

(12)

Because of the peculiar block-triangular structure of $\mathbf{K_c^L}$, the spatial NGM with small domain ($\mathbf{K_c}$, accounting only for $E_i$) is simply $\mathbf{K_c^1}$ (see again Diekmann et al.[59]). The control reproduction number can thus be found as the spectral radius of the NGM (with either large or small domain), i.e.,

$$
\mathcal{R}_c = \rho(\mathbf{K_c^L}) = \rho(\mathbf{K_c}) = \rho(\mathbf{G_c^P} + \mathbf{G_c^I} + \mathbf{G_c^A}),
$$

(13)

where

$$
\mathbf{G_c^P} = \delta^E\theta_c^P(\phi_c^E\phi_c^P)^{-1}
$$

$$
\mathbf{G_c^I} = \sigma\delta^E\delta^P\theta_c^I(\phi_c^E\phi_c^P\phi_c^I)^{-1}
$$

$$
\mathbf{G_c^A} = (1-\sigma)\delta^E\delta^P\theta_c^A(\phi_c^E\phi_c^P\phi_c^A)^{-1}
$$

(14)

are three spatially explicit generation matrices describing the contributions of post-latent infectious people, infectious symptomatic people, and asymptomatic/paucisymptomatic infectious people to the next generation of infections in a neighborhood of the DFE in the presence of disease-containment measures.

In the absence of controls, i.e., if the isolation rates $\chi_i^X$ ($X \in \{E, P, I, A\}$), the transmission reductions $\epsilon_i$, and the travel restrictions $\xi_{ij}$ are equal to zero for all $i$'s and $j$'s, then the control reproduction number $\mathcal{R}_c$ reduces to the basic reproduction number $\mathcal{R}_0$, defined as the average number of secondary infections produced by one infected individual in a population that is completely susceptible to the disease and where no containment measures are in place. $\mathcal{R}_0$ can be evaluated as the spectral radius of matrix $\mathbf{G^P} + \mathbf{G^I} + \mathbf{G^A}$, where

$$
\mathbf{G^P} = \delta^E\theta^P(\phi^E\phi^P)^{-1}
$$

$$
\mathbf{G^I} = \sigma\delta^E\delta^P\theta^I(\phi^E\phi^P\phi^I)^{-1}
$$

$$
\mathbf{G^A} = (1-\sigma)\delta^E\delta^P\theta^A(\phi^E\phi^P\phi^A)^{-1}.
$$

(15)

In the previous set of expressions, $\phi^X$ ($X \in \{E, P, I, A\}$) are diagonal matrices whose non-zero elements are $\mu + \delta^E$ (for $\phi^E$), $\mu + \delta^P$ (for $\phi^P$), $\mu + \alpha + \eta + \gamma^I$ (for $\phi^I$), and $\mu + \gamma^A$ (for $\phi^A$), while matrices $\theta^X$ ($X \in \{P, I, A\}$) are given by $\mathbf{N}\mathbf{M}^S\beta^X(\Delta)^{-1}(\mathbf{M}^X)^T$, with $\mathbf{M}^X = [M_{ij}^X]$ ($X \in \{S, P, I, A\}$) and $M_{ij}^X = M_{c,ij}^X$ evaluated with $\xi_{ij} = 0$ for all $i$'s and $j$'s, and $\Delta$ is a diagonal matrix whose non-zero entries are the elements of vector $\mathbf{u}\mathbf{N}\mathbf{M}^S$.

### Derivation of basic and control epidemicity indices.
The concept of epidemicity[26] extends previous work[24,25] where a reactivity index was defined and applied to study the transient dynamics of ecological systems characterized by steady-state behavior. To explain, in physical terms, the meaning of reactivity and of the Hermitian matrix used to derive it, consider a linear system $d\mathbf{x}/dt = \mathbf{A}\mathbf{x}$, where $\mathbf{x} = (x_1, \ldots, x_n)^T$ is the state vector and $\mathbf{A}$ is a $n \times n$ real state matrix. The system is subject to pulse perturbations $\mathbf{x}(0) = \mathbf{x}_0 > 0$. Reactivity is defined as the gradient of the Euclidean norm $||\mathbf{x}|| = \sqrt{x_1^2 + \cdots + x_n^2} = \sqrt{\mathbf{x}^T\mathbf{x}}$ of the state vector, evaluated for the fastest-growing initial perturbation, and corresponds to the spectral abscissa $\Lambda_{max}^{Re}(\cdot)$ of the Hermitian part $(\mathbf{A} + \mathbf{A}^T)/2$ of matrix $\mathbf{A}$[24]. Following Mari et al.[25], an asymptotically stable equilibrium is characterized by positive generalized reactivity if there exist small perturbations that can lead to a transient growth in the Euclidean norm of a suitable system output $\mathbf{y} = \mathbf{W}\mathbf{x}$, with matrix $\mathbf{W}$ describing a linear transformation of the system state.

In epidemiological applications, $\mathbf{W}$ should include the variables of the infection subsystem[26]. Therefore, a suitable output transformation for the problem at hand is

$$
\mathbf{W} =
\begin{bmatrix}
\mathbf{0} & w^E\mathbf{I} & \mathbf{0} & \mathbf{0} & \mathbf{0} & \mathbf{0} & \mathbf{0} & \mathbf{0} & \mathbf{0} & \mathbf{0} \\
\mathbf{0} & \mathbf{0} & w^P\mathbf{I} & \mathbf{0} & \mathbf{0} & \mathbf{0} & \mathbf{0} & \mathbf{0} & \mathbf{0} & \mathbf{0} \\
\mathbf{0} & \mathbf{0} & \mathbf{0} & w^I\mathbf{I} & \mathbf{0} & \mathbf{0} & \mathbf{0} & \mathbf{0} & \mathbf{0} & \mathbf{0} \\
\mathbf{0} & \mathbf{0} & \mathbf{0} & \mathbf{0} & w^A\mathbf{I} & \mathbf{0} & \mathbf{0} & \mathbf{0} & \mathbf{0} & \mathbf{0}
\end{bmatrix},
$$

(16)

where $w^E$, $w^P$, $w^I$, $w^A$ are the weights assigned to the variables of the infection subsystem in the output $\mathbf{y} = [w^E E_1, \ldots, w^E E_n, w^P P_1, \ldots, w^P P_n, w^I I_1, \ldots, w^I I_n, w^A A_1, \ldots, w^A A_n]^T$. Generalized reactivity for the DFE of system (Eq. (3)) is positive if the spectral abscissa of a suitable Hermitian matrix (either $\mathbf{H_0}$ or $\mathbf{H_c}$, depending on whether the spread of disease is uncontrolled or some containment measures are in place) is also positive. In SEPIAR, the expressions of matrices $\mathbf{H_0}$ and $\mathbf{H_c}$ are far from trivial, as shown below, and the evaluation of spectral abscissae typically requires numerical techniques. Note also that, since recovered individuals are not accounted for in the system output, including waning immunity would not alter the epidemicity properties of the DFE.

Let us consider the most general case of disease-containment measures being in place (which includes as a limit case also uncontrolled pathogen spread). If we note

that $\ker(\mathbf{W}) = \ker(\mathbf{W}\mathbf{J_c})$, with $\mathbf{J_c}$ being the Jacobian of SEPIAR at the DFE in the presence of controls, matrix $\mathbf{H_c}$ can be defined[25,27] as the Hermitian part of $\mathbf{W}\mathbf{J_c}(\mathbf{W})^+$, i.e.,

$$\mathbf{H_c} = H(\mathbf{W}\mathbf{J_c}(\mathbf{W})^+) = \frac{1}{2}\left\{ \mathbf{W}\mathbf{J_c}(\mathbf{W})^+ + [(\mathbf{W})^+]^T (\mathbf{J_c})^T (\mathbf{W})^T \right\}, \quad (17)$$

where $(\mathbf{W})^+$ is the right pseudo-inverse (a generalization of the concept of inverse for non-square matrices) of $\mathbf{W}$, and can be evaluated as

$$(\mathbf{W})^+ = (\mathbf{W})^T [\mathbf{W}(\mathbf{W})^T]^{-1}. \quad (18)$$

Matrix

$$\mathbf{H_c} = \begin{bmatrix} -\boldsymbol{\phi_c^E} & \frac{w^P}{2w^E}\delta^E\mathbf{I} + \frac{w^E}{2w^P}\boldsymbol{\theta_c^P} & \frac{w^E}{2w^I}\boldsymbol{\theta_c^I} & \frac{w^E}{2w^A}\boldsymbol{\theta_c^A} \\ \frac{w^P}{2w^E}\delta^E\mathbf{I} + \frac{w^E}{2w^P}\boldsymbol{\theta_c^P} & -\boldsymbol{\phi_c^P} & \frac{w^I}{2w^P}\sigma\delta^P\mathbf{I} & \frac{w^A}{2w^P}(1-\sigma)\delta^P\mathbf{I} \\ \frac{w^E}{2w^I}\boldsymbol{\theta_c^I} & \frac{w^I}{2w^P}\sigma\delta^P\mathbf{I} & -\boldsymbol{\phi_c^I} & \mathbf{0} \\ \frac{w^E}{2w^A}\boldsymbol{\theta_c^A} & \frac{w^A}{2w^P}(1-\sigma)\delta^P\mathbf{I} & \mathbf{0} & -\boldsymbol{\phi_c^A} \end{bmatrix} \quad (19)$$

is Hermitian, hence real and symmetric. Therefore all eigenvalues are real and the spectral abscissa $e_c = \Lambda_{max}^{Re}(\mathbf{H_c})$ coincides with the largest eigenvalue, which corresponds to the fastest-growing perturbation in the system output. Thus, $e_c$ can be interpreted as a control epidemicity index: if $e_c > 0$, there must exist some small perturbations to the DFE that are temporarily amplified in the system output, thus generating a transient, subthreshold epidemic wave.

Absent any containment measures, the control epidemicity index, $e_c$, reduces to the basic epidemicity index, $e_0 = \Lambda_{max}^{Re}(\mathbf{H_0})$, where

$$\mathbf{H_0} = H(\mathbf{W}\mathbf{J_0}(\mathbf{W})^+) = \frac{1}{2}\left\{ \mathbf{W}\mathbf{J_0}(\mathbf{W})^+ + [(\mathbf{W})^+]^T (\mathbf{J_0})^T (\mathbf{W})^T \right\} \quad (20)$$

and the Jacobian matrix $\mathbf{J_0}$ can be obtained from $\mathbf{J_c}$ by setting equal to zero the isolation rates $\chi_i^X$ ($X \in \{E, P, I, A\}$), the transmission reductions $\epsilon_i$, and the travel restrictions $\xi_{ij}$ for all $i$'s and $j$'s.

### The effective reproduction number and the effective epidemicity index.
The reproduction numbers and the epidemicity indices defined above can be rigorously applied only to characterize the spread of disease in a fully naïve population ($S_i = N_i \, \forall \, i$). As soon as the pathogen begins to circulate within the population, the state of the system gradually departs from the DFE. Under these circumstances, it is customary[19,21] to define a time-dependent, effective reproduction number, $\mathcal{R}(t)$, to track the number of secondary infections caused by a single infectious individual in a population in which the pool of susceptible individuals is progressively depleted, and control measures are possibly in place[58]. Similarly, it is possible to define an effective epidemicity index, $e(t)$, to evaluate the likelihood that transient epidemic waves may occur even if $\mathcal{R}(t) < 1$.

The definition of these time-dependent metrics requires to update the expressions of the spatially explicit infection matrices $\boldsymbol{\theta_c^X}$ ($X \in \{P, I, A\}$) in a time-varying fashion, i.e.,

$$\boldsymbol{\theta_c^X}(t) = \mathbf{S}(t)\mathbf{M_c^S}(t)[\mathbf{I} - \boldsymbol{\epsilon}(t)]\boldsymbol{\beta^X}[\boldsymbol{\Delta_c}(t)]^{-1}[\mathbf{M_c^X}(t)]^T, \quad (21)$$

where $\boldsymbol{\epsilon}(t)$ is a diagonal matrix whose non-zero elements represent the reduction of local transmission rates at time $t$, $\epsilon_i(t)$, $\boldsymbol{\Delta_c}(t) = \text{diag}(\mathbf{u}\sum_{X \in \{S,E,P,I,A,R\}} \mathbf{X}(t)\mathbf{M_c^X}(t))$, $\mathbf{M_c^X}(t)$ are spatially explicit contact matrices including time-varying travel restrictions, $\xi_{ij}(t)$, and $\mathbf{S}(t)$, $\mathbf{E}(t)$, $\mathbf{P}(t)$, $\mathbf{I}(t)$, $\mathbf{A}(t)$, and $\mathbf{R}(t)$ are diagonal matrices whose non-zero elements are the time-varying abundances of susceptible ($S_i(t)$), exposed ($E_i(t)$), post-latent ($p_i(t)$), symptomatic ($I_i(t)$), asymptomatic ($A_i(t)$), and recovered ($R_i(t)$) individuals in each community $i = 1...n$. The evaluation of $\mathcal{R}(t)$ and $e(t)$ also mandates an update of the transition matrices $\boldsymbol{\phi_c^X}$ to include time-dependent testing and isolation rates, $\chi_i^X$ ($X \in \{E, P, I, A\}$).

Computing the $\boldsymbol{\theta_c^X}$ and $\boldsymbol{\phi_c^X}$ matrices thus requires updating the state variables and epidemiological parameters of SEPIAR, i.e., to numerically solve Eq. (3) as the epidemic progresses and control strategies are put in place. The definition of time-varying transmission and transition terms gives rise to time-varying Jacobians and NGMs. In turn, the use of these matrices in the computation of reproduction numbers and epidemicity indices allows the evaluation of the time-varying threshold quantities $\mathcal{R}(t)$ and $e(t)$. Clearly, this type of argument works best if the depletion of the susceptible pool is relatively slow, i.e., if the initial perturbation is sufficiently small and the divergence of the epidemic trajectory from the DFE is not too large[60,61].

### Reporting summary.
Further information on research design is available in the Nature Research Reporting Summary linked to this article.

## Data availability
Secondary data were obtained from a variety of publicly accessible sources. The resident population at the provincial level is available at http://dati.istat.it/Index.aspx?QueryId=18460. The data to quantify nation-wide human mobility prior to the pandemic are available at https://www.istat.it/archivio/139381. Airport traffic data are available at http://assaeroporti.com/statistiche_201912. Surveillance data are available at https://github.com/pcm-dpc/COVID-19. All the necessary data to evaluate the basic reproduction numbers and the epidemicity indices for the Italian case study are available at https://github.com/COVID-19-routes/epidemicity-paper[62].

## Code availability
All numerical analyses have been performed with MATLAB R2020b. The code to evaluate the basic reproduction numbers and the epidemicity indices for the Italian case study are available at https://github.com/COVID-19-routes/epidemicity-paper[62].

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

## Acknowledgements

A.R., E.B., and D.P. acknowledge funding from Fondazione Cassa di Risparmio di Padova e Rovigo (IT) through its Grant 55722. A.R. acknowledges funding from the Swiss National Science Foundation via the project "Optimal control of intervention strategies for waterborne disease epidemics" (200021-172578).

## Author contributions

L.M., A.R., and M.G. designed the study. L.M. was responsible for numerical simulations. E.B. and D.P. performed parameter estimation. L.M., R.C., and S.M. collected data and finalized the analysis. L.M., R.C., E.B., D.P., S.M., A.R, and M.G. interpreted the findings and wrote the manuscript.

## Competing interests

The authors declare no competing interests.
