## [Peer Review File · Nature Communications]

Reviewers' Comments:

Reviewer #1:

Remarks to the Author:

The paper presents an epidemicity index to detect epidemic peaks near the threshold of $R=1$. The paper shows that unless the epidemicity index $e_0 < 0$ the existence of epidemic peaks or rebounds cannot be ruled out. Epidemic peaks may either delay the exponential decay of the epidemic when $R < 1$ or may grow to fully developed outbreaks.

The approach to the epidemic index presents certain analogies to a former paper by Sugihara (Nature Communications (2019) 10:2374). The justification is similar: incidence contains the information necessary to detect peaks or outbreaks. There exist several indicators that have been developed to try to predict by several days, or ideally several weeks of anticipation, the atypical growth of incidence that could trigger an outbreak (eg, Sugihara op.cit; Coelho, F.C., et al 2015. Sci. Rep. 5; Nishiura, H. et al 2009. J. R. Soc. Interface page rsif20090153; Ferguson, N.M., et al. 2016 Science 353, pp. 353-354; Angulo et al. Epidemics 24 (2018) 98-104). Some of these indicators use environmental clues and other external quantifiable quantities to inform the particular index. The idea is that a change in the index may give enough time to take preventive actions against that likely outbreak. The epidemicity index developed in this work incorporates population mobility as a key factor. This is a new important component in the development of this kind of indices. One concern, however, is that e_0 seems rather insensitive to changes in the incidence. In this example, as in many other regions of the world, R has been fluctuating around the value of 1 for several weeks, even months with these fluctuations taking values above and below 1. e_0 , in the example, remains consistently above 0 (in the illustrative case when R is below threshold.) This constancy is not informative if one wants to have an estimate of a likely time for the onset of an outbreak. As the example shows, e_0 informs the likelihood or risk of having a rebound but gives no clue as when it could occur. This fact limits its applicability. It is of course, important to have at least some information of the risk of flare ups but ideally they should be able to also inform on the (approximate) timing of the flare ups. In a related issue, the underlying assumption of the epidemicity index e_0 is that the environment is constant and that it has no influence on the dynamics of the disease. Moreover, for SARS-CoV-2 in particular, the possibility of reinfections has to be considered.

Therefore, three questions arise from the reading of this study:

1. Respiratory infections are, once established in a population, seasonal diseases and this variability is intrinsically linked to epidemic episodes. In fact, the effective constant rate is time-dependent. How essential (or inessential) is this consideration for the computation of e_0 from data.
2. What are the consequences of introducing into the model temporal immunity? Temporal immunity contributes to the endemic establishment of a disease. How is e_0 affected particularly in its sensitivity to flare ups?
3. Comparison with other approaches designed for short-time outbreak predictions (such as the ones mentioned above) need to be discussed to have a better evaluation of the performance of e_0 .

Reviewer #2:

Remarks to the Author:

The authors introduce an indicator called 'epidemicity' as an additional tool to monitor an epidemic and predict its expected evolution. This indicator is used to determine sub-threshold epidemics (i.e. when the reproductive number $R < 1$, and so the epidemic is expected to decay) that can lead to a temporary increase of cases. It is a threshold-type indicator, as R , but with a threshold at zero.

The authors use COVID-19 pandemic as an example to present (1) analytically the possible epidemic situations in different phases ($R < 1$, $e < 0$; $R < 1$, $e > 0$; $R > 1$, $e > 1$), and (2) computationally to show the effect of COVID-19 below threshold ($R < 1$) but with epidemicity larger than zero,

through numerical simulations. The computational model is then used to describe the epidemic in Italy and assess the evolution in time of the two parameters. They show that during summer 2020, when R was below 1 in Italy, the epidemicity was positive. This result is used to state that it was the reason for the successive increase of cases.

I find the introduction of this indicator intriguing and interesting in a theoretical perspective, through the authors overstate the utility of this indicator in a real-time situation, i.e. in outbreak response. Results do not support the conclusions that $e > 0$ was the factor explaining the increase of cases after summer in Italy. The introduction and explanation of epidemicity in the main text is quite poor, and it is not possible to follow the paper without resorting to the details of the Methods section. Several assumptions on the model are not explained, nor described. I have concerns on the definition of the metapopulation model (how mobility and social distancing are fit over time to the epidemic in Italy; the force of infection is not explained and seems to integrate second order transmission, when it should not).

In summary, I think it is an interesting indicator with possibly great potential from a theoretical point of view, if the technical concerns are solved. Its use as a predictive tool in outbreak response is yet to be determined. The manuscript would need a major revision in the presentation of the approach in the main text, it is really hard to follow in the present state, and it damages the interesting work.

More in detail:

1) The authors mention the epidemicity indicator for the first time in the abstract, but without giving a definition of it. We only know that it is the 'dominant eigenvalue of a matrix describing a suitable infection subsystem'. This is clearly not enough, we don't know what this index does beyond being an important tool. This is technically and phenomenologically vague. Later in the text we learn from the references that this indicator has already been introduced, but in the realm of vector-borne diseases and water-borne diseases. COVID-19 not being water-borne, it is unclear whether the epidemicity can be defined exclusively for mediated transmitted diseases or directly transmitted diseases. This is again a very important point, but it is never addressed. The authors indicate some requirements to "properly" estimate e , including a metapopulation model (though it is never called it as such, please refer to the usual notation of the definition of epidemic models), contacts between hosts, droplets and viral loads. Why would mobility be needed to define R and e ? Why contacts (I imagine, explicit contacts)? R can be easily computed on homogeneous mixing models, without mobility nor contacts. And why droplets and viral loads are needed? Anyway, these are mentioned but then never used. These do not seem to be requirements, as the authors put it, but rather choices driven by a possible objective to study the spatial diffusion of COVID-19 pdm in Italy. But the objective of the paper is the study of e as an indicator. So, at the end, the reader wonders whether e can't be computed on a simple homogeneous mixing model too, as R .

2) The focus of this study revolves around the phenomenon of extinction in a stochastic setting, which is very well studied in the literature of mathematical epidemiology. However, the authors never put in relation their theory with this existing literature. It is important to make this relation, as it would also highlight the differences and enrich the work.

3) Many of the aspects of the method presented in the main text are not clear or poorly described. Eq (1) is introduced with many parameters, but for example the force of infection is not reported. " Φ^X are the diagonal contact matrices from the infection subsystem". We don't know what this is. Many matrices are undefined or defined in an unclear way that do not allow the reader to even grasp the key elements behind the theory. Generalized reactivity is undefined. λ_{\max} are the largest eigenvalue? Unclear; and ρ was defined before as the largest eigenvalue.

4) Metapopulation model. The metapopulation model should be described in its main details in the main text. It is mentioned that the model integrates changing mobility, but this is only obtained

through an input parameter of variation. This is OK for the analytical understanding of the model. The same happens for the reduction of contacts, But how is mobility and social distancing defined when the model is fit to the epidemic curve? It should describe the changes induced by lockdown and the progressive exit from it, typically from data. This is not mentioned. The authors mention only that they use commuting data as mobility between patches, but this is clearly not applicable to 2020 given the dramatic changes induced by the implemented control measures.

5) Force of infection. This should be defined in the main text, is the key ingredient defining a model, the reader is unable to follow without this information. In addition, figure legends report the parameters β which are undefined in the main text. Overall, I have some concerns on the expression of the force of infection of the metapopulation model. There are several theoretical frameworks to define it. Here it seems from the formula that susceptibles in patch i are infected not only by infections in patches in j , neighbors of i (as for a typical metapopulation model), but also by infections in patches k neighbors of j . This means that at each time step the susceptible in i experiences a force of infection from his first and second neighbors. But this is not correct, it is a second order correlation in just one timestep. For a given transmission rate, this formulation of the force of infection gives a more rapid epidemic than the classic formulation. It is unclear why the authors need to include the second order, it is completely unnecessary and not correct.

6) Finally, the importance of epidemicity. The authors stress that it is a predictive tool for the epidemic fate in that it captures the possible temporary increase of cases, possible even when the system is below threshold. While this aspect is already accounted for by the theory of stochastic extinctions and localized clusters for subthreshold epidemics, it is not anyway the (only possible) direct cause of the increase of cases observed in Italy after summer. Many additional factors played an important role, among the most important ones: the relaxation of barrier measures by the population, the start of the activities in September after the holidays, a rapid decrease in temperature in October forcing people indoor, an inefficient control through test-trace-isolate. In addition, the number of cases that are brought by $e > 0$ are anyway negligible: we're talking of about 800 cumulated cases at national level (Fig3). The importance of this indicator should be toned down.

The epidemicity index of recurrent SARS-CoV-2 infections (ms id NCOMMS-20-43589A-Z)

Lorenzo Mari, Renato Casagrandi, Enrico Bertuzzo, Damiano Pasetto, Stefano Miccoli, Andrea Rinaldo, Marino Gatto

Response to the reviewers

We thank the editors for giving us the opportunity to revise our manuscript, and the reviewers for their careful assessment of our work. We wish to thank both reviewers also for their constructive criticism, which helped us expand the breadth of our presentation and touch on some important issues related to the transmission of SARS-CoV-2. We think that all the concerns expressed by the reviewers have been addressed fully. Detailed point-by-point replies to the reviewers' comments follow. Please note that, while the comments have been reported in full, they may have been slightly rearranged according to thematic points. The literature mentioned by the reviewers has been cross-referenced with the "References" section below. The page and line numbers in the answers below refer to the 'clean' version of the revised manuscript. A version in which changes are highlighted is also provided.

Reviewer #1

Reviewer Point P 1.1 — *The paper presents an epidemicity index to detect epidemic peaks near the threshold of $R = 1$. The paper shows that unless the epidemicity index $e_0 < 0$ the existence of epidemic peaks or rebounds cannot be ruled out. Epidemic peaks may either delay the exponential decay of the epidemic when $R < 1$ or may grow to fully developed outbreaks. The approach to the epidemic index presents certain analogies to a former paper by Sugihara (Nature Communications (2019) 10:2374)¹. The justification is similar: incidence contains the information necessary to detect peaks or outbreaks. There exist several indicators that have been developed to try to predict by several days, or ideally several weeks of anticipation, the atypical growth of incidence that could trigger an outbreak (eg, Sugihara op.cit¹; Coelho, F.C., et al 2015. Sci. Rep. 5²; Nishiura, H. et al 2009. J. R. Soc. Interface page rsif20090153³; Ferguson, N.M., et al. 2016 Science 353, pp. 353-354⁴; Angulo et al. Epidemics 24 (2018) 98–104⁵). Some of these indicators use environmental clues and other external quantifiable quantities to inform the particular index. The idea is that a change in the index may give enough time to take preventive actions against that likely outbreak. [...] Comparison with other approaches designed for short-time outbreak predictions (such as the ones mentioned above) need to be discussed to have a better evaluation of the performance of e_0 .*

Reply: We agree that there may be points of contact between epidemicity analysis and the other methods referenced by the reviewer. Specifically, the aim of both our approach and the others is similar, in that they all try to anticipate whether epidemic outbreaks (in epidemiological applications) can occur. However, there exist remarkable methodological differences between our approach and those mentioned by this reviewer. Specifically, independently of the specific goal of each paper, such as interpreting^{3,5} or anticipating^{1,2,4} the time course of an epidemic, all the methods referenced above are based on estimates of disease incidence, possibly from imperfect surveillance data. Conversely, the epidemicity analysis proposed in our work is neither based on, nor influenced by, the incidence of infection (if not indirectly, in some cases; see below).

For the sake of clarity, let us just recall that three different pairs of endemicity/epidemicity indices have been introduced in the present work, namely:

- the *basic* reproduction number and epidemicity index (\mathcal{R}_0 and e_0);
- the *control* reproduction number and epidemicity index (\mathcal{R}_c and e_c);
- the *effective* reproduction number and epidemicity index ($\mathcal{R}(t)$ and $e(t)$).

The basic and control reproduction number and epidemicity index are structural quantities, meaning that in an ordinary differential equation (ODE) model they are functions only of the model parameters and, crucially in our spatially explicit metapopulation framework, of the underlying connectivity structure (see point P 1.2 below). Therefore, they are not and cannot be influenced by the incidence of infection (unlike all methods referred to by this reviewer). Incidence necessarily refers to an observed or simulated epidemic trajectory, which in turn depends not only on the transmission parameters but also on other factors like e.g. the initial localization and magnitude of the outbreak. By contrast, the effective reproduction number and epidemicity index may be influenced (although indirectly) by disease incidence. Indeed, there exist two factors that affect these effective indices, namely: the epidemiological parameters regulating disease transmission (like in the case of the basic/control quantities), and/or the size of the susceptible compartment. The former may depend on the implementation of control measures, and thus vary over time, while the latter can be depleted if disease transmission is not effectively contained (and the infection confers at least temporary immunity; see also our answer to point P 1.5 below).

The depletion of the susceptible pool is evidently linked to disease incidence and its temporal dynamics: the higher the number of new infections, the faster the susceptible compartment gets depleted. However, our application of the SEPIAR model to the first wave of the COVID-19 pandemic in Italy suggests that the effective reproduction number and epidemicity index have been changing mostly because of the application of tight containment measures, and only marginally by the depletion of the susceptible pool. Such depletion was likely minimal in many Italian regions during the analyzed period (see point P 1.3 below for a more in-depth discussion of the parameterization of the model leading to this result).

The methodological differences between the epidemicity analysis presented here and existing incidence-based methods for short-time outbreak prediction have now been briefly described in the revised version of the manuscript (page 3, line 41). The suggested references have also been added to provide a broader context to our paper, and we thank this reviewer for bringing this interesting point to our attention.

Reviewer Point P 1.2 — *The endemicity index developed in this work incorporates population mobility as a key factor. This is a new important component in the development of this kind of indices.*

Reply: We thank this reviewer for their appreciation of the novel way in which human mobility has been accounted for in our derivation of endemicity/epidemicity indices for the transmission of SARS-CoV-2.

Reviewer Point P 1.3 — *One concern, however, is that e_0 seems rather insensitive to changes in the incidence. In this example, as in many other regions of the world, R has been fluctuating around de value of 1 for several weeks, even months with these fluctuations taking values above and below 1. e_0 , in the example, remains consistently above 0 (in the illustrative case when R is below threshold.) This constancy is not informative if one wants to have an estimate of a likely time for the onset of an outbreak. As the example shows, e_0 informs the likelihood or risk of having a rebound but gives no clue as when it could occur. This fact limits its applicability. It is of course, important to have at least some information of the risk of flare ups but ideally they should be able to also inform on the (approximate) timing of the flare ups.*

Reply: As discussed in our answer to comment P 1.1 above, the basic and control reproduction number and epidemicity index are structural quantities that cannot be influenced by the incidence of infection. By contrast, the effective reproduction number ($\mathcal{R}(t)$) and the epidemicity index ($e(t)$), which—we believe—are the quantities the reviewer is here referring to, can vary over time and may indeed be influenced (albeit indirectly) by disease incidence.

Concerning the application of our spatially explicit SEPIAR model to the first wave of COVID-19 transmission in Italy (Figure 5 of the original manuscript, now Figure 6), it is evident that $\mathcal{R}(t)$ and $e(t)$ did vary over time. This variability is mostly linked to the change in the transmission parameters induced by the application of control measures. In fact, while calibrating the model on the available data, we allowed the transmission rates to take different values over different time windows, corresponding to the timing of the implementation of the main nation-wide restrictions, or their lifting⁶. Specifically, the effect of the containment measures was parameterized by assuming that the transmission parameters had a sharp decrease after the containment measures announced in Italy at the end of February and the beginning of March, and that they were further reduced in the following weeks as the country was effectively entering full lockdown. The temporal variations of $\mathcal{R}(t)$ and $e(t)$ follow closely this progressive tightening of the containment measures, echoing previous modeling results concerning the effectiveness of control measures on early transmission dynamics⁷. By contrast, only marginally did the dynamics of the susceptible compartment affect $\mathcal{R}(t)$ and $e(t)$, consistently with model-based estimates suggesting that the depletion of the susceptible pool during the first wave of the pandemic has been small to negligible in northern Italy, and basically null in the south of the country⁶. All these details are now better described in the revised manuscript (page 12, line 256; page 28, line 534).

Therefore, the ex-post calibration procedure used here to parameterize the model explains the purported inability of $\mathcal{R}(t)$ and $e(t)$ to rapidly react to (or even predict) changes in the incidence of new infections. Conversely, data-based estimates of the effective reproduction number, like those produced in Italy by Istituto Superiore di Sanità, and incidence-based projections of epidemic dynamics, like those referred to by this reviewer in comment P 1.1 above, are continuously updated and may thus follow more closely the temporal dynamics of the outbreak. A possible linkage

between these different approaches could be represented by data assimilation schemes (e.g. Ensemble Kalman Filter), in which state and model parameters are jointly and repeatedly re-assessed over time. We have recently used these techniques to analyze cholera transmission in Haiti⁸ in conjunction with spatially explicit metapopulation models not unlike SEPIAR (albeit representing different underlying epidemiological dynamics). Joint, frequent updates of parameter values and state variables may allow, on one hand, to reliably evaluate the effective reproduction number and epidemicity index, as well as their temporal dynamics; and, on the other, to produce reasonable epidemiological projections with a lead time of at least a few weeks. An integrative approach like the one just outlined here would represent a very important extension to our modeling framework—one that would merit a separate manuscript by itself, though. We believe this point to be important, therefore we have added a comment in the revised version of the manuscript (page 15, line 323). We thank this reviewer for suggesting another relevant direction for our discussion.

May we also reinforce that the importance of our epidemicity index is related to its integrating power for the current prognostic use of the effective reproduction number $\mathcal{R}(t)$. It is customary for policymakers in Italy (and in most other countries) to relax restriction measures when $\mathcal{R}(t) < 1$, a comprehensible index for the general public. However, here we show that this is a necessary but not sufficient condition to avoid sub-threshold epidemics possibly coalescing into clusters of outbreaks. Sufficiency only comes when $e(t) < 0$, and in this sense the epidemicity index may represent a more meaningful indicator of the epidemiological state.

Reviewer Point P 1.4 — *In a related issue, the underlying assumption of the epidemicity index e_0 is that the environment is constant and that it has no influence on the dynamics of the disease. [...] Respiratory infections are, once established in a population, seasonal diseases and this variability is intrinsically linked to epidemic episodes. In fact, the effective constant rate is time-dependent. How essential (or inessential) is this consideration for the computation of e_0 from data.*

Reply: This reviewer is right. We did not include seasonal effects in the epidemiological parameters when evaluating basic/control reproduction numbers and epidemicity indices, although we are certainly aware of the possibility that the transmission of SARS-CoV-2 might follow seasonal patterns similarly to what other respiratory infections do, with inherent complexities like e.g. in the case of influenza⁹. We did not account for seasonality here for a twofold reason.

First, epidemicity analysis by construction gives necessary conditions for the possible existence of transient (typically short-term) epidemic outbreaks triggered by a pulse perturbation to a stable epidemiological steady state. The first wave of the COVID-19 pandemic started in Italy during winter, and the calibration of the SEPIAR model used to run most of the analyses discussed in the present work does indeed include transmission rates that cannot help but refer to that specific season. Actually, we could not have done anything different in this regard, given the emerging nature of the pathogen. For the same reason, we could not look into endemic dynamics, thus we only limited our attention to the epidemicity properties of the disease-free equilibrium (DFE). This is now explicitly mentioned in the revised manuscript (page 5, line 95). Note that epidemicity analysis can be applied to endemic equilibria as well (although in that case obtaining analytical results may be overwhelming), namely to establish whether short-term epidemic dynamics is possible on top of endemic transmission¹⁰.

Second, a proper inclusion of seasonality in the evaluation of basic/control reproduction numbers requires a considerably more involved mathematical framework than the one used to build next-generation matrices, which is essentially based on linear stability analysis. Indeed, to evaluate reproduction numbers in environments where seasonal fluctuations are approximately periodic, one can resort to Floquet theory¹¹. The application of this theory to local, well-mixed models is relatively straightforward^{12–15}. However, its extension to spatially explicit metapopulation models proves quite computationally demanding¹⁶. Things are even more complicated if we turn our attention to basic/control epidemicity indices. In fact, the study of transient dynamics in periodically forced systems is still in its infancy, and concepts like instantaneous vs. period reactivity^{17,18} (which, suitably generalized, would likely represent the foundation of epidemicity analysis in seasonal environments) are yet to be fully developed.

All this being said, we note that in our computations of the effective reproduction number and epidemicity index for the Italian case study (which, we think, is what this reviewer is referring to when mentioning a "time-dependent effective constant rate") we did include possible seasonal effects, albeit in an approximate, empirical manner. In fact, the evaluation of $\mathcal{R}(t)$ and $e(t)$ relies on a time-varying parameterization of the SEPIAR model in which the transmission rates were allowed to take different values over different time windows (see also the answer to comment P 1.3 above). While time-varying transmission rates were included primarily to account for the epidemiological effects of the progressive tightening of containment measures, their seasonal variations during the calibration window of the model would be indirectly captured as well. Therefore, to address the reviewer's question, frequent updates to the estimation of the model parameters (like e.g. in the data assimilation framework envisioned in the reply to comment P 1.3) may represent at present our best shot at a data-informed inference of any seasonality-induced effects on the effective reproduction number and epidemicity index, while a more solid theoretical approach still needs to be developed. While we do not see this as a limitation of the present study—due to the emerging nature of the pathogen, as mentioned above—we agree with this reviewer that seasonality may become a key component of future modeling efforts aimed at studying post-pandemic SARS-CoV-2 transmission dynamics, i.e. if/when the pathogen establishes as endemic¹⁹. A comment on seasonal transmission has been added to the revised version of the Methods section (page 28, line 544).

Reviewer Point P 1.5 — *Moreover, for SARS-CoV-2 in particular, the possibility of reinfections has to be considered. [...] What are the consequences of introducing into the model temporal immunity? Temporal immunity contributes to the endemic establishment of a disease. How is e_0 affected particularly in its sensitivity to flare ups?*

Reply: This is another interesting point. We did not consider immunity loss because the consensus is that acquired immunity following SARS-CoV-2 infection lasts at least a few months, and we were chiefly interested in the early phases of disease spread when the whole population is susceptible and the problem of re-infections, if present at all, is certainly negligible. However, it is quite straightforward to show that adding immunity loss to our model does not change the basic/control reproduction numbers and epidemicity indices defined in the text. A brief proof follows.

Let us modify the equations of SEPIAR to include the loss of acquired immunity. The only two equations that require an update are those for the dynamics of susceptible (S_i) and recovered

individuals (R_i), which respectively become

$$\dot{S}_i = \mu(N_i - S_i) - \lambda_i S_i + \zeta R_i$$

and

$$\dot{R}_i = \gamma^I(I_i + I_i^h) + \gamma^A(A_i + A_i^q) - (\mu + \zeta)R_i,$$

where ζ is the rate at which recovered individuals lose their acquired immunity, and all other variables and parameters retain their original meaning. The Jacobian matrix that describes the dynamics of the system in a neighborhood of the DFE in the presence of controls (which represents the most general setting) thus becomes

$$\mathbf{J}_c^\dagger = \begin{bmatrix} -\mu\mathbf{I} & \mathbf{0} & -\theta_c^P & -\theta_c^I & -\theta_c^A & \mathbf{0} & \mathbf{0} & \mathbf{0} & \mathbf{0} & \zeta\mathbf{I} \\ \mathbf{0} & -\phi_c^E & \theta_c^P & \theta_c^I & \theta_c^A & \mathbf{0} & \mathbf{0} & \mathbf{0} & \mathbf{0} & \mathbf{0} \\ \mathbf{0} & \delta^E\mathbf{I} & -\phi_c^P & \mathbf{0} & \mathbf{0} & \mathbf{0} & \mathbf{0} & \mathbf{0} & \mathbf{0} & \mathbf{0} \\ \mathbf{0} & \mathbf{0} & \sigma\delta^P\mathbf{I} & -\phi_c^I & \mathbf{0} & \mathbf{0} & \mathbf{0} & \mathbf{0} & \mathbf{0} & \mathbf{0} \\ \mathbf{0} & \mathbf{0} & (1-\sigma)\delta^P\mathbf{I} & \mathbf{0} & -\phi_c^A & \mathbf{0} & \mathbf{0} & \mathbf{0} & \mathbf{0} & \mathbf{0} \\ \mathbf{0} & \chi^E & \mathbf{0} & \mathbf{0} & \mathbf{0} & -(\mu + \delta^E)\mathbf{I} & \mathbf{0} & \mathbf{0} & \mathbf{0} & \mathbf{0} \\ \mathbf{0} & \mathbf{0} & \chi^P & \mathbf{0} & \mathbf{0} & \delta^E\mathbf{I} & -(\mu + \delta^P)\mathbf{I} & \mathbf{0} & \mathbf{0} & \mathbf{0} \\ \mathbf{0} & \mathbf{0} & \mathbf{0} & \eta\mathbf{I} + \chi^I & \mathbf{0} & \mathbf{0} & \sigma\delta^P\mathbf{I} & -(\mu + \alpha + \gamma^I)\mathbf{I} & \mathbf{0} & \mathbf{0} \\ \mathbf{0} & \mathbf{0} & \mathbf{0} & \mathbf{0} & \chi^A & \mathbf{0} & (1-\sigma)\delta^P\mathbf{I} & \mathbf{0} & -(\mu + \gamma^A)\mathbf{I} & \mathbf{0} \\ \mathbf{0} & \mathbf{0} & \mathbf{0} & \gamma^I\mathbf{I} & \gamma^A\mathbf{I} & \mathbf{0} & \mathbf{0} & \gamma^I\mathbf{I} & \gamma^A\mathbf{I} & -(\mu + \zeta)\mathbf{I} \end{bmatrix}.$$

Matrix \mathbf{J}_c^\dagger retains the same spectral properties as the original Jacobian matrix \mathbf{J}_c . It thus follows that the dynamics of the infection subsystem around the DFE may still be described by the reduced-order Jacobian \mathbf{J}_c^\star . As a result, the definition of the next-generation matrices remains unchanged, and so does the derivation of the basic/control reproduction numbers. As for epidemicity, since the recovered class is not involved in the output transformation, the basic/control epidemicity indices for the DFE remain unchanged too.

On the contrary, the evaluation of the effective reproduction number and epidemicity index may be quantitatively affected by the introduction of vanishing immunity. In this case, in fact, the transmission terms contained in the time-varying Jacobian matrix must include updated values of the state variables, all of which are either directly ($S_i(t)$, $R_i(t)$) or indirectly (the others) touched by the modification of the model. However, given the relatively short time scales considered in our analyses, at least with respect to the typical estimates of the duration of acquired immunity (a few months, at least), the results obtained considering immunity loss do not change much with respect to the scenario in which acquired immunity is permanent (as considered in the manuscript), in terms of both model simulations⁶ and the estimation of the effective reproduction number and epidemicity index.

As an example, Figure R1 below shows that even considering a very fast loss of acquired immunity ($\zeta = 1/30 \text{ days}^{-1}$, corresponding to an average immune period of about one month) leads to estimates that are almost indistinguishable from those obtained with permanent immunity ($\zeta = 0$, as in the manuscript) in terms of hospital admissions (panel a), effective reproduction number (b), and effective epidemicity index (c). Note that the introduction of a fast immunity loss does indeed alter the dynamics of disease transmission, namely by allowing a prompt replenishment

of the susceptible pool (panel d). However, because of the relative low level of susceptible depletion, this qualitative change is too small to alter the epidemic trajectory and the estimates of $\mathcal{R}(t)$ and $e(t)$.

We have now revised the manuscript in the light of this insightful remark (page 31, line 598; page 34, line 675; page 38, line 744).

Reviewer #2

Reviewer Comment — *The authors introduce an indicator called ‘epidemicity’ as an additional tool to monitor an epidemic and predict its expected evolution. This indicator is used to determine sub-threshold epidemics (i.e. when the reproductive number $R < 1$, and so the epidemic is expected to decay) that can lead to a temporary increase of cases. It is a threshold-type indicator, as R , but with a threshold at zero.*

The authors use COVID-19 pandemic as an example to present (1) analytically the possible epidemic situations in different phases ($R < 1, e < 0$; $R < 1, e > 0$; $R > 1, e > 1$), and (2) computationally to show the effect of COVID-19 below threshold ($R < 1$) but with epidemicity larger than zero, through numerical simulations. The computational model is then used to describe the epidemic in Italy and assess the evolution in time of the two parameters. They show that during summer 2020, when R was below 1 in Italy, the epidemicity was positive. This result is used to state that it was the reason for the successive increase of cases.

I find the introduction of this indicator intriguing and interesting in a theoretical perspective, through the authors overstate the utility of this indicator in a real-time situation, i.e. in outbreak response. Results do not support the conclusions that $e > 0$ was the factor explaining the increase of cases after summer in Italy. The introduction and explanation of epidemicity in the main text is quite poor, and it is not possible to follow the paper without resorting to the details of the Methods section. Several assumptions on the model are not explained, nor described. I have concerns on the definition of the metapopulation model (how mobility and social distancing are fit over time to the epidemic in Italy; the force of infection is not explained and seems to integrate second order transmission, when it should not).

In summary, I think it is an interesting indicator with possibly great potential from a theoretical point of view, if the technical concerns are solved. Its use as a predictive tool in outbreak response is yet to be determined. The manuscript would need a major revision in the presentation of the approach in the main text, it is really hard to follow in the present state, and it damages the interesting work.

Reply: We thank this reviewer for their appreciation of the theoretical implications of our work, as well as for their constructive criticism. The detailed comments by this reviewer helped us clarify several technical aspects of the manuscript and, hopefully, highlight some more practical implications of the epidemicity index and our work in general. Point-by-point replies to the reviewer’s comments follow.

Reviewer Point P2.1 — *The authors mention the epidemicity indicator for the first time in the abstract, but without giving a definition of it. We only know that it is the ‘dominant eigenvalue of a matrix describing a suitable infection subsystem’. This is clearly not enough, we don’t know what this index does beyond being an important tool. This is technically and phenomenologically vague.*

Reply: We recognize we were indeed a bit too vague in the abstract about the definition of the epidemicity index. We have now clarified its nature as a threshold-type quantity (page 1, line 9)

Figure R1: **The effect of immunity loss on estimates of the effective reproduction number $\mathcal{R}(t)$ and epidemicity index $e(t)$ for the first wave of the COVID-19 pandemic in Italy.** (a) Hospital admissions at the country scale. Green empty dots represent the curated data that has been used⁶ to calibrate the SEPIAR model assuming permanent immunity. The black line and shading are the median and the 95% confidence interval, respectively, of 2,000 simulations with parameter values drawn from a posterior distribution estimated from calibration on data. The SEPIAR model has been run until the end of July for validation. The cyan dashed line refers to the median of other 2,000 simulations with parameter values drawn from the same posterior distribution but accounting for fast immunity loss ($\zeta = 1/30 \text{ days}^{-1}$). (b) Temporal dynamics of the effective reproduction number obtained from SEPIAR (red curve and shading: median and 95% confidence interval for simulations with permanent immunity; cyan dashed curve: median for simulations with immunity loss). (c) Effective epidemicity index computed from SEPIAR (blue curve and shading: median and 95% confidence interval for simulations with permanent immunity; cyan dashed curve: median for simulations with immunity loss). (d) Country-wide fraction of susceptible people (black curve and shading: median and 95% confidence interval for simulations with permanent immunity; cyan dashed curve: median for simulations with immunity loss). All other parameters and simulation details as in Figure 6 in the manuscript.

and remarked what can happen if the epidemicity index is positive (page 1, line 11). Thanks for raising this important point.

Reviewer Point P 2.2 — *Later in the text we learn from the references that this indicator has already been introduced, but in the realm of vector-borne diseases and water-borne diseases. COVID-19 not being water-borne, it is unclear whether the epidemicity can be defined exclusively for mediated transmitted diseases or directly transmitted diseases. This is again a very important point, but it is never addressed.*

Reply: We thank this reviewer for highlighting a matter that improperly we gave for granted. As noted, in the past, we have applied epidemicity analysis mainly to water-borne and vector-borne diseases^{10,20}, but the definition of an epidemicity index does not require those peculiar modes of transmission. Indeed, epidemicity indices can be defined and evaluated for any disease, route(s) of transmission, and contact network (irrespective of its complexity and spatial structure), provided that disease transmission is described through a compartmental model defined as a system of coupled ODEs. Clearly, epidemicity indices are crucially disease-specific. Indeed, the ones we propose here are based on a state-of-the-art representation of COVID-19 transmission in space and time (the SEPIAR model^{6,7}).

For context, the concept of epidemicity originates from ecological studies aimed at estimating the resilience of ecosystems at equilibrium. In most cases, such estimates describe the rate at which the effect of perturbations to a stable steady state are dampened by the system dynamics only in the very long run, as time goes to infinity (asymptotic stability), and are based on the eigenvalues of the system evaluated at the equilibrium. In some cases, though, the resilience of ecological systems has been characterized also in terms of their transient response to external perturbations. For biological systems, this short-term analysis was first proposed in a seminal work²¹ that introduced the concept of reactivity, using earlier results from a theory of generalized stability born in the context of meteorology and fluid dynamics²². The physical meaning of reactivity refers to whether, in the short term, impulsive perturbations to a stable steady state can grow significantly before they decay. Evidently, in this case, eigenvalues would provide no information about the transient behavior of the system. The epidemicity index proposed here actually represents an epidemiological implementation of generalized reactivity analysis¹⁴, an extension of the concept of reactivity that we elaborated a few years ago. Thus, not only is epidemicity analysis a widely applicable tool in epidemiology, but also the generalized reactivity framework in which it is rooted is completely general and finds potential applications in many different fields.

We have modified the manuscript to better emphasize the generality of epidemicity analysis (page 2, line 35) and frame it in the larger context of generalized reactivity analysis (page 2, line 32; page 13, line 279; page 37, line 722), and we thank this reviewer for pointing this out.

Reviewer Point P 2.3 — *The authors indicate some requirements to “properly” estimate e , including a metapopulation model (though it is never called it as such, please refer to the usual notation of the definition of epidemic models), contacts between hosts, droplets and viral loads. Why would mobility be needed to define R and e ? Why contacts (I imagine, explicit contacts)? R can be easily computed on homogeneous mixing models, without mobility nor contacts. And why droplets and viral loads are needed? Anyway, these are mentioned but then never used. These do not seem to be requirements, as the authors put it, but rather choices driven by a possible objective to study the spatial diffusion of COVID-19 pdm in Italy. But the objective*

of the paper is the study of e as an indicator. So, at the end, the reader wonders whether e can't be computed on a simple homogeneous mixing model too, as R .

Reply: This reviewer is right, in that the sentence they are referring to was probably unclear. What we meant—and we stand by—is that in a spatially explicit setting, a natural framework to describe epidemic dynamics unfolding e.g. over a whole country, or any other realistic landscape, accounting for spatial connectivity and the related mobility fluxes is crucial. Previous research has in fact shown that estimates of the reproduction number evaluated at local (i.e. via homogeneous-mixing models) vs. spatial scales (e.g. via metapopulation models) may diverge^{23,24}. Clearly, the connectivity pathways to be included in a metapopulation model vary according to the spatial scale of interest. That is why we mentioned human mobility and droplets as possible examples of large-vs. small-scale connectivity, and retained the former in our country-scale model.

We have revised the manuscript to clarify this important point (page 3, line 50). We have also resorted to the term "metapopulation" to refer to our spatially explicit model, as suggested by this reviewer (e.g. page 3, line 57; page 13, line 276). As for the possibility to evaluate an epidemicity index for a homogeneous-mixing model, please see also our reply to point P 2.2 above.

Reviewer Point P 2.4 — *The focus of this study revolves around the phenomenon of extinction in a stochastic setting, which is very well studied in the literature of mathematical epidemiology. However, the authors never put in relation their theory with this existing literature. It is important to make this relation, as it would also highlight the differences and enrich the work.*

Reply: Frankly, we would not characterize our study as being focused on epidemic extinction. In fact, while we indeed are chiefly interested in sub-threshold outbreaks, i.e. epidemic dynamics occurring with \mathcal{R}_0 (or $\mathcal{R}_c, \mathcal{R}(t) < 1$), our aim is that of investigating the conditions leading to possible short-term outbreaks or the revamping of transmission, rather than the eventual disappearance thereof. In a sense, we aim at complementing the current use of reproduction numbers—widespread even in public communication of the state of the epidemic—highlighting the possible existence, and the associated dangers, of sub-threshold outbreaks potentially coalescing into a major revamping of the number of infections. In fact, our work explores the structural conditions that may be conducive to further epidemic waves even when $\mathcal{R}_c < 1$, i.e. when in the public eye the epidemic is being contained and transmission should supposedly be receding.

That being said, the reviewer is certainly right in pointing out that there exists a large body of literature on stochastic epidemic dynamics, including epidemic extinction. Research efforts, often built upon foundational work by Bartlett in the 1950s and 1960s, have been devoted to a variety of issues, like the study of the probability of epidemic development²⁵ or stochasticity-induced disappearance^{26–29}, and the expected time to epidemic extinction³⁰. However, studies like these invariably refer to the case $\mathcal{R}_0 > 1$ (or to the "barely subcritical"³¹ regime $\mathcal{R}_0 \approx 1$), under the assumption that an epidemic would shrink exponentially for $\mathcal{R}_0 < 1$. In this respect, some affinities may exist between our work and the literature on so-called stuttering transmission chains³², which are typical of pathogens spreading inefficiently in a population. In this case too, though, the estimation of quantities like the total size of an outbreak is done under the hypothesis that the average number of cases (evaluated over different observations of the process) declines monotonically over time when $\mathcal{R}_0 < 1$ ^{33,34}, and the possibility of sub-threshold, yet non-negligible

outbreaks is left uninvestigated.

While revising the Discussion section of the manuscript, we paid attention to highlight similarities and differences between our approach and the literature on stuttering transmission chains (page 13, line 283) and, more in general, stochastic epidemics (page 13, line 289). We thank this reviewer for bringing up this interesting point.

Reviewer Point P2.5 — *Many of the aspects of the method presented in the main text are not clear or poorly described. Eq (1) is introduced with many parameters, but for example the force of infection is not reported. “ Φ^X are the diagonal contact matrices from the infection subsystem”. We don’t know what this is. Many matrices are undefined or defined in an unclear way that do not allow the reader to even grasp the key elements behind the theory. Generalized reactivity is undefined. Λ_{\max} are the largest eigenvalue? Unclear; and ρ was defined before as the largest eigenvalue.*

Reply: We acknowledge that a better organization of the technical material was in order. We have now revised the Methods section of the manuscript to present the methodological aspects of our work in a hopefully more accessible manner. The related changes, we believe, greatly improved the presentation of our methodology, and we thank this reviewer for their important feedback. Answers to the specific points made by this reviewer follow:

- the force of infection was reported right after the description of model (3). We have anticipated a reference to it in the paragraph right below the model (page 30, line 578);
- the infection subsystem was referred to in many points of both Methods and supplements, but was never defined in plain words. This is now done in the revised version of the Methods section (page 34, line 673);
- we have double-checked that all matrices (and all mathematical quantities, in general) are briefly defined in the main text (page 4, line 81) and fully explained in the Methods section. To that end, we had to move the derivation of the basic/control reproduction numbers, as well as of the effective reproduction number and epidemicity index, from the Supplementary Information to the Methods (page 33, line 647; page 39, line 769). To keep the length of this section within reasonable limits, we moved the catalog of dynamical behaviors of SEPIAR (Figure 6, now 2) from the Methods to the main text (page 6, line 122).
- we have briefly defined what generalized reactivity is (page 2, line 30);
- concerning $\rho(\cdot)$ vs. $\Lambda_{\max}(\cdot)$, we needed different notations to distinguish between spectral radius and spectral abscissa. Referring for clarity to the general case of a complex $N \times N$ matrix $\mathbf{A} \in \mathbb{C}^{N \times N}$ with eigenspectrum (the set of all its complex eigenvalues) $K \equiv \{\lambda_i\}$, the spectral *radius*, $\rho(\mathbf{A})$, is defined as the maximum *absolute* value of its eigenvalues, i.e.

$$\rho(\mathbf{A}) = \max_{\lambda_i \in K} |\lambda_i|.$$

Therefore $\rho(\mathbf{A})$ is a nonnegative real number, and the condition $\rho(\mathbf{A}) < 1$ implies that all the eigenvalues are inside the unit circle on the complex plane. Similarly, the spectral *abscissa*,

$\alpha(\mathbf{A})$, is the maximum *real* part of the spectrum, i.e.

$$\alpha(\mathbf{A}) = \max_{\lambda_i \in K} \operatorname{Re} \lambda_i,$$

so that $\alpha(\mathbf{A})$ is a real number $-\infty < \alpha(\mathbf{A}) < \infty$. The condition $\alpha(\mathbf{A}) < 0$ implies that all the eigenvalues are in the left half of the complex plane. In general, $\rho \geq |\alpha|$. Unfortunately, while $\rho(\cdot)$ is universally recognized as the standard notation for the spectral radius, the less common concept of spectral abscissa is present in the literature with different notations, such as $\alpha(\cdot)$, $\eta(\cdot)$, $\lambda_{\max}^{\operatorname{Re}}(\cdot)$. In our manuscript, all these lowercase greek letters are already used to indicate model parameters (α : disease-induced mortality; η : hospitalization rate; λ : force of infection). Therefore, we introduced $\Lambda_{\max}(\cdot)$ for the eigenvalue of maximal real part, i.e.

$$\Lambda_{\max}(\mathbf{A}) = \lambda \quad \text{with} \quad \lambda \in K \quad \text{and} \quad \operatorname{Re} \lambda \geq \operatorname{Re} \lambda_i, \quad \forall \lambda_i \in K.$$

This eigenvalue of maximal real part (if unique) is sometimes called the *dominant* eigenvalue. In the present case, matrix \mathbf{A} is real and symmetric (hence Hermitian), thus its spectrum is real, its dominant eigenvalue unique (albeit possibly with algebraic multiplicity greater than one) and coincident with the spectral radius, i.e.

$$\Lambda_{\max}(\mathbf{A}) = \alpha(\mathbf{A}).$$

Of course, the relation

$$|\Lambda_{\max}(\mathbf{A})| \leq \rho(\mathbf{A})$$

still holds, so that this is a different concept from the spectral radius. We agree that this choice of notation could be misleading. Therefore, in the revised manuscript we now use $\Lambda_{\max}^{\operatorname{Re}}(\cdot)$ for the spectral abscissa (that is we capitalize $\lambda_{\max}^{\operatorname{Re}}(\cdot)$ to distinguish it from the force of infection λ) and define epidemicity in terms of spectral abscissa, dropping the concept of “dominant eigenvalue” (see page 5, line 102).

Reviewer Point P2.6 — *Metapopulation model. The metapopulation model should be described in its main details in the main text. It is mentioned that the model integrates changing mobility, but this is only obtained through an input parameter of variation. This is OK for the analytical understanding of the model. The same happens for the reduction of contacts, But how is mobility and social distancing defined when the model is fit to the epidemic curve? It should describe the changes induced by lockdown and the progressive exit from it, typically from data. This is not mentioned. The authors mention only that they use commuting data as mobility between patches, but this is clearly not applicable to 2020 given the dramatic changes induced by the implemented control measures.*

Reply: The mathematical structure of the metapopulation model was already described in the Methods section of the main text, together with a summary of its application to the Italian case study, which is fully documented in published material elsewhere^{6,7}. As already explained in our reply to

point P 1.3 above, for most of the analyses presented in the manuscript we used a parameterization of the model that refers to the earliest phase of the epidemic, in which SARS-CoV-2 was spreading basically unnoticed in the population. In that case, we used a business-as-usual mobility scenario, on top of which we explored the possible effects of a preventive planning of travel restrictions, described by parameter ξ_{ij} . While in several instances we have considered spatially homogeneous controls ($\xi_{ij} = \xi$, see Figures 3, 4, and 5), our representation of travel restrictions is flexible enough to account for local/regional/country-wide travel bans (Figure 6; supplementary Figures S4, S5, S6, S7, S8, and S9).

A different, yet obviously related problem is that of assessing the actual effects of the COVID-19 pandemic on human mobility. In this respect, this reviewer is certainly right in suggesting that we could not use a business-as-usual mobility scenario for the later phases of the pandemic, when disease spread was dramatically apparent. Indeed, in the time-varying parameterization⁶ of the model used for estimating the effective reproductive number and epidemicity index for the Italian case study (Figure 5 in our previous submission, now Figure 6 in the revised version), not only the transmission rates (see again point P 1.3 above) but also the mobility patterns were allowed to change over time, reflecting the impacts of both travel restrictions imposed by the authorities and behavioral change induced by the awareness of the risks associated with mobility. Specifically, between-province mobility was progressively reduced as the epidemic unfolded according to estimates obtained through mobility data from mobile applications^{35,36}.

We have significantly revised the paragraph in the Methods devoted to the parameterization of the model and its fit to the Italian epidemic curve, making sure to include all of the details highlighted by the reviewer (page 27, line 525), including how human mobility was described in the model to account for the effects of the pandemic and the associated containment measures (page 28, line 550). We also introduced a couple of comments in the main text to better clarify this important point (page 7, line 153; page 12, line 256).

Reviewer Point P 2.7 — *Force of infection. This should be defined in the main text, is the key ingredient defining a model, the reader is unable to follow without this information. In addition, figure legends report the parameters β which are undefined in the main text.*

Reply: Please note that the force of infection was already defined in the manuscript, specifically in the Methods section, where the transmission parameters β^X ($X \in \{P, I, A\}$) were also defined in both the paragraph right after the model equations and Table 2. Table 2 is also referenced in all the relevant figure captions. Anyway, we took the reviewer's comment as an opportunity to double-check that the definitions of all mathematical terms are readily accessible in the revised version of the manuscript.

Reviewer Point P 2.8 — *Overall, I have some concerns on the expression of the force of infection of the metapopulation model. There are several theoretical frameworks to define it. Here it seems from the formula that susceptibles in patch i are infected not only by infections in patches in j , neighbors of i (as for a typical metapopulation model), but also by infections in patches k neighbors of j . This means that at each time step the susceptible in i experiences a force of infection from his first and second neighbors. But this is not correct, it is a second order correlation in just one timestep. For a given transmission rate, this formulation of the force*

of infection gives a more rapid epidemic than the classic formulation. It is unclear why the authors need to include the second order, it is completely unnecessary and not correct.

Reply: A crucial point that must be clarified to let the reader better understand our definition of the force of infection is that, in our metapopulation model, human mobility does not actually entail the relocation of individuals. Rather, it mostly depicts daily commuting flows (as already pointed out in the manuscript), which are described in our metapopulation model by the instantaneous spatial-mixing matrices ($M_{c,ij}^X$, with $X \in \{S, E, P, I, A, R\}$, for the mobility fluxes between i and j). However, the epidemiologically-relevant contacts between the residents of two different communities may not necessarily occur in either community; in fact, they could happen anywhere else, say in community k , between residents of i and j simultaneously traveling to k . Our definition of the force of infection does actually formalize these complex, mobility-related contact processes—and neglecting this factor would lead to an underestimation of the potential for the virus to spread geographically.

A practical example may perhaps help here. Let us consider New York City. Many residents of, say, Queens commute to Manhattan on a daily basis for work or study. There, they will find not only people who are resident in that borough, but also people who live elsewhere, say in Brooklyn, and who are also commuting for study or work to Manhattan. As a result, the total contacts between the residents of Queens and Brooklyn include not only those occurring because of the direct mobility fluxes between these two boroughs, but also those resulting from mobility to another borough (Manhattan, in this example) that attracts commuting fluxes from both of them. In epidemiological terms, a susceptible resident of Brooklyn commuting to Manhattan may be exposed to the virus because of social mixing with infectious people who either reside in Manhattan or are traveling there from another borough (or from anywhere else within the spatial domain of interest, for that matter). Our formulation aims at capturing this mixing behavior and does not include an incorrect "second neighbour" effect, as interpreted by this reviewer.

This important aspect of our definition of the force of infection was addressed in the original manuscript only through its mathematical formulation, while a much-needed plain-word explanation was missing. We thank this reviewer for pointing this out, which helped us clarify a crucial feature of our model. We have thoroughly revised for clarity the paragraph of the Methods section devoted to the description of the force of infection (page 31, line 601).

Reviewer Point P2.9 — *Finally, the importance of epidemicity. The authors stress that it is a predictive tool for the epidemic fate in that it captures the possible temporary increase of cases, possible even when the system is below threshold. While this aspect is already accounted for by the theory of stochastic extinctions and localized clusters for subthreshold epidemics, it is not anyway the (only possible) direct cause of the increase of cases observed in Italy after summer. Many additional factors played an important role, among the most important ones: the relaxation of barrier measures by the population, the start of the activities in September after the holidays, a rapid decrease in temperature in October forcing people indoor, an inefficient control through test-trace-isolate.*

Reply: We do not intend epidemicity as a truly "predictive" tool, and we did not mean to convey that message in the previous version of the manuscript. Given its nature of necessary (yet not sufficient) condition for transient, sub-threshold outbreaks to occur, a positive epidemicity index to

us represents more of a cautionary tale about the fact that, although the conditions for long-term endemicity are not met, short-term outbreaks may be expected, provided that the system is excited in a suitable way (see also our response to point P 2.2 above).

Concerning the specific application to the Italian case study, we definitely agree with this reviewer that a positive epidemicity index was not the only direct cause of the recrudescence of SARS-CoV-2 transmission in the Fall. Indeed, it was not even an indirect cause of the observed surge of new infections, and we apologize for some occasional unfortunate choice of words (like in the first paragraph of the Discussion section of the original version of the manuscript, now revised). Rather, it indicated that a rebound of the epidemic could not have been ruled out, and this suggestion alone might have advised in favor of more stringent control measures—despite local and country-wide reproduction numbers being less than one. We believe we were quite cautious in the original manuscript not to state that the positivity of the epidemicity index was the only culprit of the so-called second wave of transmission in Italy, but we have double-checked throughout the manuscript that no ambiguities might arise and further emphasized this important point (e.g. page 12, line 271; page 14, line 310; page 14, line 319).

As per the observation that "many additional factors played an important role" we must remark that our approach allows us to take them fully into account if Bayesian parameter calibration is adequately repeated over time (or a suitable data assimilation scheme is in place)—as we did in fact for the first wave of the pandemic in Italy (see our reply to point P 1.3 above). Thus the effective reproduction number $\mathcal{R}(t)$ and the effective epidemicity indices $e(t)$ would fully account also for the additional factors mentioned by this reviewer.

In conclusion, we see $e(t)$ not as a further factor that adds to other causes in determining outbreak risk, but rather as a synthetic index that depends upon the epidemiological parameters and their variations, and that may signal the risk, at the country level, of possible sub-threshold epidemics. We have stressed this important point in the revised version of the manuscript (e.g. page 3, line 44; page 15, line 337).

Reviewer Point P 2.10 — *In addition, the number of cases that are brought by $e > 0$ are anyway negligible: we're talking of about 800 cumulated cases at national level (Fig3). The importance of this indicator should be toned down.*

Reply: We respectfully disagree. Figure 3 (now 4) displays simulation examples run for a few selected parameter combinations. In the case referred to by this reviewer, 100 individuals who are initially exposed to the virus (in one province) eventually result in around 800 total cases over a three-month-long period, that is a 700 % increment. A more systematic analysis (Figure 4, now 5) shows that using different (lower) values of the controls (still, all leading to $\mathcal{R}_c < 1$ and $e_c > 0$) can lead to higher counts, up to 2,000 cases for $\mathcal{R}_c \approx 1$. Note that this would still represent an underestimation of the total case count linked to the initial perturbation, because all simulations are evaluated over a fixed timespan of 90 days (which allows a fair comparison of different control strategies), by the end of which many simulated outbreaks may still be ongoing (as evident from the green curves in the left panels of Figure 3—now 4). We have added a couple of notes on this point in the revised version of the manuscript (page 10, line 218; page 11, line 234). We thank the reviewer for this comment that prompted us to clarify an important, quantitative aspect of our

work, but maintain that the main message of our work still stands. As far as the importance of the concept of epidemicity is concerned, please also refer to our answer to comment P 2.9 above.

Relatedly, it is also worth keeping in mind that all simulations describe outbreaks triggered by impulsive perturbations to the DFE, so as to conform with the tenet of epidemicity analysis as we have presented it. Situations in which the system is continuously excited by a "rain" of exposed individuals might lead to even higher case counts. However easy to simulate, accounting for this different type of perturbation would require a much more complicated mathematical framework, basically involving an extension of general stability analysis to non-autonomous systems²². We have thus decided not to explore this interesting avenue to avoid adding further mathematical complications to a manuscript that admittedly is already quite demanding from that perspective.

In conclusion, we wish to thank again the editors for granting us the opportunity to revise our work, and both reviewers for their constructive comments, which helped us improve the quality of the manuscript. We look forward with confidence to a, hopefully positive, editorial decision.

References

1. Rypdal, M. & Sugihara, G. Inter-outbreak stability reflects the size of the susceptible pool and forecasts magnitudes of seasonal epidemics. *Nature Communications* **10** (May 2019).
2. Coelho, F. C. & de Carvalho, L. M. Estimating the Attack Ratio of Dengue Epidemics under Time-varying Force of Infection using Aggregated Notification Data. *Scientific Reports* **5** (Dec. 2015).
3. Nishiura, H., Chowell, G., Heesterbeek, H. & Wallinga, J. The ideal reporting interval for an epidemic to objectively interpret the epidemiological time course. *Journal of The Royal Society Interface* **7**, 297–307 (July 2009).
4. Ferguson, N. M. *et al.* Countering the Zika epidemic in Latin America. *Science* **353**, 353. <http://science.sciencemag.org/content/353/6297/353.abstract> (July 2016).
5. Angulo, M. T. & Velasco-Hernandez, J. X. Robust qualitative estimation of time-varying contact rates in uncertain epidemics. *Epidemics* **24**, 98–104. ISSN: 1755-4365. <http://www.sciencedirect.com/science/article/pii/S1755436517300750> (2018).
6. Bertuzzo, E. *et al.* The geography of COVID-19 spread in Italy and implications for the relaxation of confinement measures. *Nature Communications* **11**, 4264 (Aug. 2020).
7. Gatto, M. *et al.* Spread and dynamics of the COVID-19 epidemic in Italy: Effects of emergency containment measures. *Proceedings of the National Academy of Sciences USA* **117**, 10484–10491 (Apr. 2020).
8. Pasetto, D. *et al.* Near real-time forecasting for cholera decision making in Haiti after Hurricane Matthew. *PLoS Computational Biology* **14**, e1006127 (2018).
9. Casagrandi, R., Bolzoni, L., Levin, S. A. & Andreasen, V. The SIRC model and influenza A. *Mathematical Biosciences* **200**, 152–169 (Jan. 2020).

10. Mari, L., Casagrandi, R., Rinaldo, A. & Gatto, M. Epidemicity thresholds for water-borne and water-related diseases. *Journal of theoretical biology* **447**, 126–138 (June 2018).
11. Klausmeier, C. Floquet theory: A useful tool for understanding nonequilibrium dynamics. *Theoretical Ecology* **1**, 153–161 (2008).
12. Bacaër, N. & Guernaoui, S. The epidemic threshold of vector-borne diseases with seasonality. *Journal of Mathematical Biology* **53**, 421–436 (2006).
13. Bacaër, N. & Ait Dads, E. On the biological interpretation of a definition for the parameter R_0 in periodic population models. *Journal of Mathematical Biology* **65**, 601–621 (2012).
14. Mari, L., Casagrandi, R., Rinaldo, A. & Gatto, M. A generalized definition of reactivity for ecological systems and the problem of transient species dynamics. *Methods in Ecology and Evolution* **8**, 1574–1584 (June 2017).
15. Stella, E., Mari, L., Gabrieli, J., Barbante, C. & Bertuzzo, E. Permafrost dynamics and the risk of anthrax transmission: A modelling study. *Scientific Reports* **10**, 16460 (2020).
16. Mari, L., Casagrandi, R., Bertuzzo, E., Rinaldo, A. & Gatto, M. Floquet theory for seasonal environmental forcing of spatially explicit waterborne epidemics. *Theoretical Ecology* **7**, 351–365 (2014).
17. Vesipa, R. & Ridolfi, L. Impact of seasonal forcing on reactive ecological systems. *Journal of Theoretical Biology* **419**, 23–35 (2017).
18. Lutscher, F. & Wang, X. Reactivity of communities at equilibrium and periodic orbits. *Journal of Theoretical Biology* **493**, 110240 (2020).
19. Kissler, S. M., Tedijanto, C., Goldstein, E., Grad, Y. H. & Lipsitch, M. Projecting the transmission dynamics of SARS-CoV-2 through the postpandemic period. *Science* **368**, 860–868 (Apr. 2020).
20. Mari, L., Casagrandi, R., Bertuzzo, E., Rinaldo, A. & Gatto, M. Conditions for transient epidemics of waterborne disease in spatially explicit systems. *Royal Society Open Science* **6**, 181517 (May 2019).
21. Neubert, M. G. & Caswell, H. Alternatives to resilience for measuring the responses of ecological systems to perturbations. *Ecology* **78**, 653–665 (Apr. 1997).
22. Farrell, B. F. & Ioannou, P. J. Generalized Stability Theory. Part I: Autonomous Operators. *Journal of the Atmospheric Sciences* **53**, 2025–2040 (July 1996).
23. Gatto, M. *et al.* Generalized reproduction numbers and the prediction of patterns in waterborne disease. *Proc. Natl. Acad. Sci. USA* **109**, 19703–19708 (Nov. 2012).
24. Gatto, M. *et al.* Spatially Explicit Conditions for Waterborne Pathogen Invasion. *The American Naturalist* **182**, 328–346 (Sept. 2013).
25. Van Herwaarden, O. A. & Grasman, J. Stochastic epidemics: Major outbreaks and the duration of the endemic period. *Journal of Mathematical Biology* **33**, 581–601 (1995).
26. Van Herwaarden, O. A. Stochastic epidemics: The probability of extinction of an infectious disease at the end of a major outbreak. *Journal of Mathematical Biology* **35**, 793–813 (1997).

27. Schwartz, I. B., Billings, L., Dykman, M. & Landsman, A. Predicting extinction rates in stochastic epidemic models. *Journal of Statistical Mechanics: Theory and Experiment* **2009**, P01005 (2009).
28. Allen, L. J. & Lahodny Jr, G. E. Extinction thresholds in deterministic and stochastic epidemic models. *Journal of Biological Dynamics* **6**, 590–611 (2012).
29. Bertuzzo, E., Finger, F., Mari, L., Gatto, M. & Rinaldo, A. On the probability of extinction of the Haiti cholera epidemic. *Stochastic Environmental Research and Risk Assessment* **30**, 2043–2055 (2016).
30. Nåsell, I. On the time to extinction in recurrent epidemics. *Journal of the Royal Statistical Society: Series B (Statistical Methodology)* **61**, 309–330 (1999).
31. Brightwell, G., House, T. & Luczak, M. Extinction times in the subcritical stochastic SIS logistic epidemic. *Journal of Mathematical Biology* **77**, 455–493 (2018).
32. Lloyd-Smith, J. O. *et al.* Epidemic dynamics at the human-animal interface. *Science* **326**, 1362–1367 (2009).
33. Blumberg, S. & Lloyd-Smith, J. O. Inference of R_0 and transmission heterogeneity from the size distribution of stuttering chains. *PLoS Computational Biology* **9**, e1002993 (2013).
34. Blumberg, S. & Lloyd-Smith, J. O. Comparing methods for estimating R_0 from the size distribution of subcritical transmission chains. *Epidemics* **5**, 131–145 (2013).
35. Pepe, E. *et al.* *The residual social distancing in Italy during Phase 2* tech. rep. (COVID-19 Mobility Monitoring project, 2020). <https://covid19mm.github.io/in-progress/2020/05/27/fifth-report.html> (2020).
36. Pepe, E. *et al.* COVID-19 outbreak response, a dataset to assess mobility changes in Italy following national lockdown. *Scientific Data* **7** (July 2020).

Reviewers' Comments:

Reviewer #1:

Remarks to the Author:

I have read the authors' reply. My commentaries and points have been satisfactorily addressed

Reviewer #2:

Remarks to the Author:

I would like to thank the authors for a thorough review and for fully responding to all points I raised in an efficient and overly satisfactory way. The improvements in the Methods presentation and Discussion make this work an important paper for outbreak response, I suggest this to be made rapidly available.

The epidemicity index of recurrent SARS-CoV-2 infections (ms id NCOMMS-20-43589B)

Lorenzo Mari, Renato Casagrandi, Enrico Bertuzzo, Damiano Pasetto, Stefano Miccoli, Andrea Rinaldo, Marino Gatto

Reviewer #1

Reviewer Point P 1.1 — *I have read the authors' reply. My commentaries and points have been satisfactorily addressed*

Reviewer #2

Reviewer Comment — *I would like to thank the authors for a thorough review and for fully responding to all points I raised in an efficient and overly satisfactory way. The improvements in the Methods presentation and Discussion make this work an important paper for outbreak response, I suggest this to be made rapidly available.*

We thank the editors for giving us the opportunity to revise our manuscript, and the reviewers for their appreciation of our revision. We are ready to submit the final version of our work, along with all the requested editorial checklists, and we look forward with confidence to the forthcoming editorial decision.